# Native N-glycome profiling of single cells and ng-level blood isolates using label-free capillary electrophoresis-mass spectrometry

Anne-Lise Marie [1,2], Yunfan Gao [1,2] & Alexander R. Ivanov [1] ✉

The development of reliable single-cell dispensers and substantial sensitivity improvement in mass spectrometry made proteomic profiling of individual cells achievable. Yet, there are no established methods for single-cell glycome analysis due to the inability to amplify glycans and sample losses associated with sample processing and glycan labeling. In this work, we present an integrated platform coupling online in-capillary sample processing with high-sensitivity label-free capillary electrophoresis-mass spectrometry for N-glycan profiling of single mammalian cells. Direct and unbiased quantitative characterization of single-cell surface N-glycomes are demonstrated for HeLa and U87 cells, with the detection of up to 100 N-glycans per single cell. Interestingly, N-glycome alterations are unequivocally detected at the single-cell level in HeLa and U87 cells stimulated with lipopolysaccharide. The developed workflow is also applied to the profiling of ng-level amounts (5–500 ng) of blood-derived protein, extracellular vesicle, and total plasma isolates, resulting in over 170, 220, and 370 quantitated N-glycans, respectively.

Glycans are assemblies of linear and branched monosaccharide chains that govern molecular interactions and, therefore, cell communication, signal transduction, pathogen recognition, and immune responses[1–3]. In living mammalian cells, glycosylation (catalyzed by glycosyltransferases and glycosidases) produces a highly complex and vast repertoire of cellular glycans, with a colossal structural diversity[1]. Mammalian cells are covered with a dense layer of glycans and the proteins and lipids they are attached to, termed the *glycocalyx*, which is involved in various vital cellular processes[4–6]. The type, size, structure, and charge of cell surface glycans may affect the biological properties of the cells and their susceptibility to potential viral infections[2,4,7]. Mammalian glycoconjugates, located on the cell membrane and extra- or intracellular space, play crucial roles in physiological and pathological events[2,4,5,7,8]. Alterations in the glycomic profiles of glycoproteins, e.g., overexpression of sialylated or core fucosylated glycans, or increased levels of complex-type branched glycans, may promote the acquisition of cellular features required for the malignant transformation of cells[1,9–12]. Recent studies also reported altered glycosylation patterns in patients with Alzheimer's disease[13,14].

Glycosylation abnormalities represent an overt source of potential biomarkers for the diagnostic, prognostic, and treatment monitoring of various human diseases, including autoimmune, congenital, oncological, and neurodegenerative pathologies[5,7,12]. In the emerging field of glycomedicine, novel therapeutic drugs or vaccines targeting tumor-associated carbohydrate antigens, like Lewis antigens and polysialic acids, are also currently in clinical evaluation[5,15,16].

Single-cell omics is a multi-faceted field that can potentially answer a myriad of questions in biomedical and clinical or fundamental biology applications[17–19]. The commonly used approaches of analyzing populations of thousands to millions of cells in bulk samples may not reflect the cellular heterogeneity and hide subtle cell ome variabilities. On the contrary, the analysis of single cells may result in the characterization of distinct cell subpopulations and rare cells, and in the differentiation of various cell states, which are often overlooked in bulk sample analyses[17,18]. Single-cell analysis is also beneficial to certain applications (e.g., minimally invasive liquid biopsy/micro-biopsy-based cancer diagnosis and monitoring), where the amounts of original biological material may be limited for downstream molecular

[1]Barnett Institute of Chemical and Biological Analysis, Department of Chemistry and Chemical Biology, Northeastern University, 360 Huntington Ave., Boston, MA 02115, US. [2]These authors contributed equally: Anne-Lise Marie, Yunfan Gao. ✉e-mail: a.ivanov@northeastern.edu

profiling. Single-cell omics encompasses a broad spectrum of analytical techniques that rely on vastly different technological principles, including genomics, transcriptomics, metabolomics, lipidomics, proteomics, and glycomics[17,20,21]. In contrast to single-cell genomics and transcriptomics, single-cell proteomics and glycomics remain in their infancy. The analysis of proteomes and glycomes at the single-cell level is limited by the minute amounts of starting biological material and the inability to directly amplify protein and glycan species. For single-cell proteomics, ultra-sensitive mass spectrometry (MS)-based analytical platforms are under development[22–25]. The Slavov group developed Single-Cell ProtEomics by Mass Spectrometry (SCoPE-MS) techniques that rely on liquid chromatography (LC)-MS/MS methods with tandem mass tag (TMT)-labeling of peptides and a carrier channel for quantifying protein covariation across hundreds and potentially thousands of single cells[26,27]. The Kelly[28,29] and Mechtler[30] groups succeeded in developing reliable platforms for label-free single-cell proteomics using in-house or commercial nanoliter-liquid handlers. Gebreyesus et al. implemented a fully automated workflow combining microchips and MS for sample preparation and bottom-up analysis, which resulted in the detection of >1500 protein groups from one single mammalian PC-9 cell[31]. Our group recently reported a capillary electrophoresis (CE)-MS method for top-down single-cell proteomic profiling[32]. This proof-of-concept approach allowed us to identify >60 unique proteoforms in single HeLa cells[33].

Contrary to the rapidly growing field of single-cell proteomics, single-cell glycomics (SCG, which is a new suggested abbreviation) demonstrated lagging progress. This may be explained by (i) the substantially lowered amounts of biological material available (glycosylation may account for only ~1–10% of the total mass of one human glycoprotein[1]), and (ii) the necessity to label the native glycans in the most commonly used positive electrospray ionization (ESI)-MS mode for increased ionization efficiency and detectability of released glycans, which obviously induces additional and substantial sample losses during the sample processing. Analytical platforms were implemented for total cellular glycome analysis of various mammalian cells, but these studies required large amounts of cells (~$10^5$–$10^7$ cells)[34,35]. To date, only a few analytical technologies have been developed for SCG. The Johnston group developed the SUrface-protein Glycan And RNA-seq (SUGAR-seq) method for the analysis of the transcriptome, extracellular epitopes, and N-linked glycosylation at the single-cell level, using biotinylated lectins and anti-biotin antibodies combined with multimodal RNA-seq technology[36]. Oligonucleotide-labeled lectins were used by the Tateno group for RNA-tag sequencing-based glycomic profiling of single cells, which enabled the acquisition of 39 lectin-binding signals per single-cell and provided an informative picture of cell surface glycosylation[37]. Roan and co-workers implemented a cytometry time-of-flight-lectin (CyTOF-Lec) technique for the simultaneous detection and quantification of proteins and glycans on the surface of human cells[38]. In this approach, lanthanide-conjugated lectins were combined with traditional CyTOF mass cytometry using lanthanide-conjugated antibodies to specifically bind glycans and proteins and quantify them (through the lanthanide metal quantification) using inductively-coupled-plasma-MS (ICP-MS). The developed technique showed that CD4+ T cell surface glycosylation could influence the susceptibility of CD4+ T cells to viral infection. Nevertheless, the above-described approaches involve tedious, expensive, and time-consuming analytical workflows with sophisticated instrumentation and, most importantly, do not allow the direct analysis, quantitation, and accurate structural characterization of the glycans. Furthermore, some cell surface glycans may not interact with the lectins selected in the developed methodologies. These last few years, computational modeling software tools to predict the glycome at the single-cell level were also developed, based, for example, on single-cell RNA-seq transcriptomics data[39]. Yet, so far, methods enabling the direct analysis, characterization, and quantitation of cell glycomes at the single-cell level have not been reported yet.

In this work, we present the developed in-capillary sample processing methodology coupled with high-sensitivity label-free CE-MS for native N-glycan profiling of minute amounts of biomedically relevant specimens and single mammalian cells. Blood-derived isolates (serum IgM and IgG, total plasma, and plasma extracellular vesicles (EVs)), and mammalian cells (HeLa and U87) are loaded and processed in the CE capillary for N-glycan release with PNGase F prior to CE-MS analysis. The mild conditions used for N-glycan release allow us to preserve the cell membrane integrity and specifically liberate cell surface N-glycans. As in our previous work[40], N-glycans are analyzed in their native underivatized state to preserve their endogenous glycan features and eliminate the drawbacks associated with any labeling procedures, including incomplete derivatization, side-products, sample losses during cleanup steps, and high levels of defucosylation/desialylation during sample preparation and MS analysis. For glycan analysis of intact mammalian cells (1–10 cells), the manual hydrodynamic cell loading procedure described in our previous work[32] is further optimized not only to increase the robustness and throughput of cell loading but also to improve the detectability and separation of the released N-glycans during CE-MS analysis. The label-free in-capillary sample preparation approach coupled online with CE-MS vastly simplifies the analytical workflow and eliminates sample losses associated with sample handling and transfer steps of the offline approach, in comparison to our previously published work[40] and numerous other reported techniques for offline analyses of released glycans[41–43]. The developed workflow allows us to analyze 0.1–5 ng-levels of model proteins and 5–500 pL-levels of total plasma, as well as single mammalian cells. To the best of our knowledge, such an approach enabling direct analysis and quantification of N-glycans derived from one single-cell has not been reported yet. In addition, biochemical stimulation of mammalian cells induces significant qualitative and quantitative changes in the treated cells' glycosylation profiles and confirms the potential of the method to detect cell glycome alterations in biological and biomedical applications at the single-cell level.

## Results

We recently reported the implementation of an in-capillary sample processing method coupled online with high-sensitivity CE-MS for top-down analysis of single mammalian cells and limited samples[32]. The analytical workflow we previously developed for cell loading and in-capillary sample processing was adapted in the presented here study for CE-MS analysis of cell surface N-glycans with additional improvements. Besides, the non-labeling strategy for N-glycan analysis we recently developed[40], which demonstrated a sensitivity not yet reported for N-glycan profiling of scarce amounts of blood-derived isolates (sub-0.1 nL-level), was selected and optimized for CE-MS analysis of N-glycans released from single mammalian cells after their injection into the CE capillary. Figure 1 depicts the analytical workflow we developed for N-glycan profiling of single cells as well as for limited amounts of blood-derived isolates or other biological samples. The injected samples were sandwiched between two plugs of PNGase F and incubated inside the capillary. After the digestion step, the CE and ESI-MS voltages were triggered for online CE-MS analysis of the released N-glycans in their native non-labeled state. We started with the analysis of simpler sample types, moving on to more challenging samples.

### N-glycome profiling of blood-derived isolates

**N-glycan profiling of human serum IgM.** Human immunoglobulin M (IgM), a heavily glycosylated multimeric protein ($Mr_{th}$ 970 kDa and 1080 kDa for pentameric and hexameric forms, respectively), was selected to develop and optimize the in-capillary sample processing method for N-glycan release coupled online to label-free CE-MS analysis (see Supplementary Note 1). N-glycans account for ~10 % of the

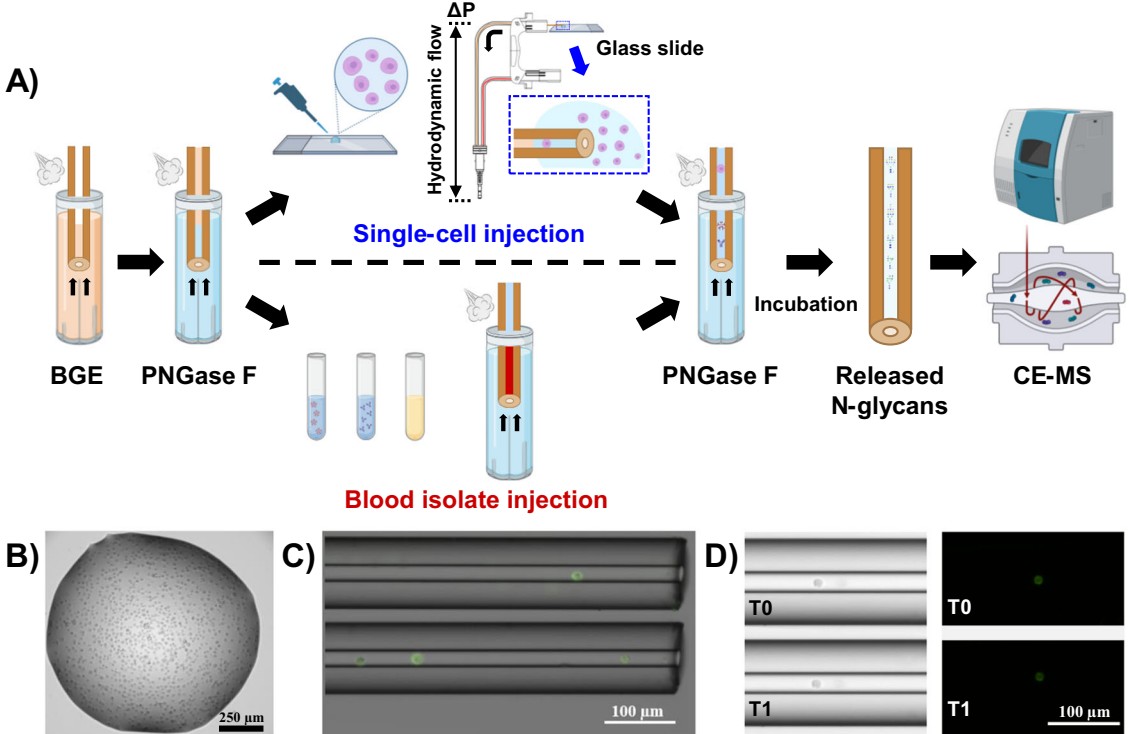

**Fig. 1 | CE-MS-based experimental workflow for N-glycan profiling of single mammalian cells and blood-derived isolates. A** Schematic representation of the analytical platform developed for in-capillary sample processing and CE-MS analysis of N-glycans released from single mammalian cells and blood-derived isolates (model serum proteins, whole plasma, and plasma-derived EVs). Individual mammalian cells are injected manually using the height difference between the inlet and outlet ends of the CE capillary. The single-cell plug is sandwiched between two plugs (1 nL each) of a PNGase F digestion solution. After 30 min (blood-derived isolates) or 1 h (mammalian cells) incubation with PNGase F, the CE, and MS electrospray voltages are triggered for label-free CE-MS analysis of released N-glycans. **B** Representative image of a HeLa cell suspension droplet used for single-cell loading ($n = 5$ technical replicates). **C** Overlay of bright-field and fluorescence images showing one single HeLa (top view) and three HeLa (bottom view) cells loaded into the CE capillary ($n = 3$ technical replicates). **D** Comparative bright-field (on the left) and fluorescence (on the right) images displaying cell integrity before (T0) and after (T1) the in-capillary deglycosylation step with PNGase F ($n = 3$ technical replicates). Some components of (**A**) were created with http://BioRender. com (publishing license: QM26MTBD9H).

total mass of IgM[44], and the serum level of IgM is in the range 0.4–2.5 mg/mL[45,46]. CE-MS analysis of IgM-derived in-capillary released N-glycans resulted in the identification of $173 \pm 6$ ($n = 3$) non-redundant N-glycan compositions in human serum IgM isolate for injected amounts of 5 ng (i.e., 5 fmol) of protein, corresponding to ~500 pg of N-glycans and equivalent to the amount of IgM isolated from ~3 nL of human serum (Fig. 2A and Supplementary Data 1). Interestingly, the injection of sample amounts as small as 0.1 ng (i.e., 100 amol) of IgM, corresponding to ~10 pg of N-glycans and equivalent to ~60 pL of human serum, resulted in the detection and identification of $132 \pm 9$ ($n = 3$) N-glycans (Fig. 2A). Considering the current developments in single-cell analysis, it is worth noting that these minute amounts of proteins and glycans (~100 pg and ~10 pg, respectively) are equivalent to the protein and glycan content of one single mammalian cell[20]. Using our developed label-free CE-MS-based workflow, the number of identified N-glycans was increased ~7-fold, compared to previously reported studies focused on N-glycan profiling of human serum IgM[47,48]. Figure 2B, C displays the fractional distributions of fucosylated and sialylated N-glycans identified in IgM, using 0.1 ng and 5 ng injected amounts of IgM. While highly fucosylated (up to 6 fucose residues) and highly sialylated (up to 6 sialic acid (SiA) residues) N-glycans were detected in the CE-MS analyses performed with either 5 ng or 0.1 ng of IgM, heptafucosylated and heavily sialylated (7–11 SiA residues) N-glycans were detected only with 5 ng of IgM injected amounts, indicating the very low abundances of these uncommon glycans.

We noticed that the developed and optimized workflow was well suited to the analysis of 5 ng and sub-ng amounts of serum IgM. Using these tiny sample amounts and intensive rinsing steps between runs, no

significant carryover derived from the analysis of preceding IgM samples was observed, based on the analysis of the water blank control sample (see "Methods" section). The injection of larger amounts of protein (e.g., 25–100 ng), which could potentially increase the glycan coverage of IgM, would require the re-optimization of several parameters, including the glycosidase: protein substrate ratio, the incubation time, and the rinsing steps between runs to efficiently clean the capillary. This scale-up workflow would obviously increase the sample processing and total analysis times. Since the goal of our study was to develop an effective and quick CE-MS-based workflow applicable to single-cell analysis, we estimated that glycan amounts released from the digestion of 0.1–5 ng of model glycoprotein within the CE capillary should reflect well the amounts of glycans released from one to ten mammalian cells.

**N-glycan profiling of human serum IgG.** We tested the developed N-glycan profiling workflow with the CE-MS analysis of human IgG ($Mr_{th}$ 150 kDa), another class of immunoglobulin less glycosylated than IgM. N-glycans account for ~2% of the total mass of IgG, and the human serum level of IgG is in the range 7–16 mg/mL[45,46]. The developed CE-MS method allowed us to identify $142 \pm 9$ ($n = 3$) non-redundant N-glycan compositions in human serum IgG isolate for injected amounts of 5 ng (i.e., 33 fmol) of protein, corresponding to ~100 pg of N-glycans and equivalent to the amount of IgG isolated from ~500 pL of serum (Fig. 2A and Supplementary Data 1). Compared to our previous studies reporting N-glycan profiling of IgG[40,49], the injected amounts of IgG were decreased ~5-fold. With such low injected amounts, the number of identified N-glycans still largely exceeded (~6-fold) the number of N-glycans reported in human serum IgG by other

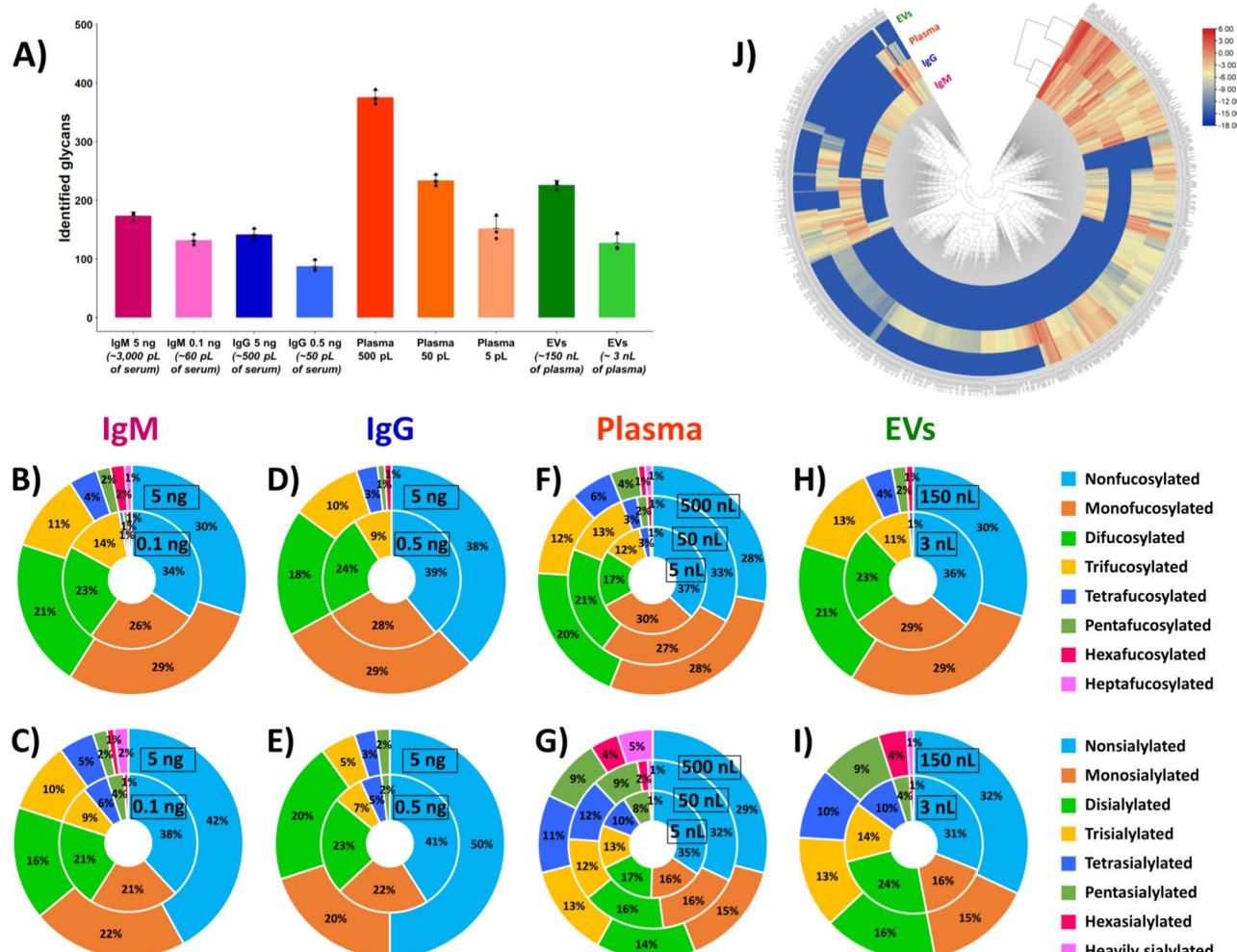

**Fig. 2 | CE-MS-based N-glycan profiling of human blood-derived isolates.**
**A** Number of N-glycans ($n$ = 3 technical replicates, data are presented as mean values ± the standard deviations (SD), black dots correspond to individual data points) identified in the four analyzed blood-derived isolates (human serum IgM, human serum IgG, total plasma, and EV isolate). **B–H** Fractional distributions of fucosylated N-glycans detected in IgM (**B**), IgG (**D**), total plasma (**F**), and EVs (**H**). **C–I** Fractional distributions of sialylated N-glycans detected in IgM (**C**), IgG (**E**), total plasma (**G**), and EVs (**I**). **J** Euclidean distance-based hierarchical clustering of quantitative glycomic profiles of IgM, IgG, total plasma, and EVs, using injected

amounts of 5 ng of IgM, 5 ng of IgG, 500 pL of plasma, and 50 nL of EV isolate (corresponding to ~150 nL of plasma), respectively. Red, yellow, and light blue colors (corresponding to $\log_2$ values of the normalized signal intensities ranging from 6 to −3, −3 to −6, and −6 to −12, respectively) display high, medium, and low relative abundances, based on the normalized N-glycan signal intensities (see "Methods" section and Supplementary Data 2). N-glycans that are not detected in the blood-derived samples are highlighted in dark blue. Source data are provided as a Source Data file.

groups[50,51]. As observed for IgM, the injection of protein amounts larger than 5 ng of IgG, in order to increase the glycan coverage of IgG, would require a workflow re-optimization. We deemed it more relevant to decrease the injected protein amount to mimic the amount of glycans released from one single-cell. The injection of ~0.5 ng (i.e., 3 fmol) of IgG resulted in the identification of 88 ± 10 ($n$ = 3) N-glycans (Fig. 2A). These scarce amounts of IgG correspond to only ~10 pg of N-glycans and isolates from ~50 pL of serum. Figure 2D, E displays the fractional distributions of fucosylated and sialylated N-glycans identified in IgG. In our previous work[40], we showed that hexa- and heavily sialylated N-glycans could not be detected in CE-MS analysis of non-labeled IgG-derived N-glycans using injected amounts equivalent to ~25 ng of IgG. As expected, glycans with a degree of sialylation ≥6 were not detected in such low 0.5–5 ng IgG sample amounts using the presented here workflow, since even larger injected amounts of serum IgG also did not allow us to detect them previously.

**N-glycan profiling of total human plasma.** Our developed workflow allowed us to analyze sub-nL volumes of total human plasma isolate

and enabled direct online N-glycan profiling of plasma volumes as small as 5 pL, which was not reported before. Data processing of CE-MS analyses resulted in the identification of 375 ± 12, 234 ± 10, and 152 ± 21 ($n$ = 3) non-redundant N-glycan compositions in whole blood plasma for injected amounts of 500 pL, 50 pL, and 5 pL of plasma (i.e., ~1500, 150, and 15 pL of human blood), respectively (Fig. 2A and Supplementary Data 1). In our previous work[40], 210 ± 12, and 62 ± 31 N-glycans were identified in whole plasma for injected amounts equivalent to ~160 pL and ~80 pL of plasma, respectively, using label-free CE-MS analysis of plasma-derived glycans released offline. The presented here results, therefore, demonstrate the superior performance of the developed in-capillary sample processing-based workflow for straightforward and unbiased N-glycome analysis of minute amounts of physiological fluids. As shown in Fig. 2F, G, hexa- and heptafucosylated, and heavily sialylated (i.e., ≥7 SiA residues) N-glycans were not detected using ~5 pL of plasma injected volumes. These highly fucosylated and sialylated N-glycans were only detected using larger plasma volumes. Glycans containing up to 13–14 SiA residues were detected with the injection of 50 and 500 pL of plasma.

**N-glycan profiling of blood-derived extracellular vesicles.** The developed in-capillary workflow was applied to the analysis of N-glycans released from human plasma-derived extracellular vesicles (EVs), another attractive source of disease biomarkers[52,53]. Experiments were carried out with the injection of a purified EV isolate, containing ~1 × 10⁴ EV particles/nL (see "Methods" section). CE-MS analysis resulted in the detection and identification of 127 ± 14 and 226 ± 7 N-glycans in the total EV isolate, using 1 nL and 50 nL of EV isolate injection volumes, respectively (containing approximately 1 × 10⁴ EVs and 5 × 10⁵ EVs, respectively) (Fig. 2A, Supplementary Data 1, and Supplementary Note 1). These injected amounts are equivalent to the EV content of ~3 nL and ~150 nL of plasma, respectively. Figure 2H, I displays the fractional distributions of fucosylated N-glycans detected in the total EV isolate using injection volumes of 1 nL and 50 nL of the EV isolate (i.e., ~3 nL and ~150 nL of plasma equivalents). Interestingly, the injection of volumes as small as 1 nL of EV isolate resulted in the detection of tetrafucosylated and hexasialylated N-glycans. The injection of larger volumes of EV isolate (i.e., 50 nL) allowed us to increase the coverage of fucosylated glycans with the detection of penta- and hexafucosylated N-glycans, which were not detected using 1 nL of EV isolate. The injection of 50 nL of EV isolate also resulted in the detection of heavily sialylated N-glycans (up to 14 SiA residues), undetectable using 1 nL volume of EV isolate. Compared to our previous study[40], similar coverage of EV-derived N-glycans was achieved using the developed label-free CE-MS technique, based on the number and types (i.e., degrees of fucosylation and sialylation) of identified N-glycans for similar injected amounts. Finally, the presented here CE-MS-based workflow (similarly to our previously reported label-free CE-MS method[40]) allowed us to further expand the catalog of blood-derived EV N-glycans, compared to other studies reported for N-glycome profiling of biofluid-derived EVs[49,54], using injected amounts as low as ~150 nL of plasma (i.e., ~400 nL of blood).

**Differential N-glycan profiling of IgM, IgG, whole plasma, and EV isolates from blood.** A qualitative and quantitative comparative analysis of N-glycans detected in the four types of analyzed blood-derived isolates (IgM, IgG, total plasma, and EVs) was conducted with an exhaustive list of 679 glycans, encompassing all the non-redundant N-glycan compositions identified in the four blood isolates. This differential analysis further demonstrated the uniqueness and high complexity of the four examined N-glycomes (Fig. 2J and Supplementary Data 2). Interestingly, and as expected, IgM and IgG immunoglobulins and total plasma and plasma-derived EVs were clustered in two distinct clades, based on their respective N-glycome profiles (Fig. 2J). 68 N-glycans were uniquely detected in human serum IgM (Supplementary Fig. 1A), among which 25% were highly fucosylated (i.e., 5−7 fucose residues) and 10% were highly sialylated (i.e., 5−11 SiA residues) N-glycans. 185 N-glycans were uniquely detected in total human plasma (Supplementary Fig. 1A), among which 12% were highly fucosylated (i.e., 5−7 fucose residues) and 31% were highly sialylated (i.e., 5−14 SiA residues) N-glycans. In contrast, the numbers of N-glycans uniquely detected in human serum IgG and total EV isolate were relatively low (13 and 20 N-glycans, respectively, Supplementary Fig. 1A). As expected, based on our previous studies, unique IgG N-glycans did not exhibit high degrees of fucosylation and sialylation since, as described above, highly fucosylated and sialylated glycans were not detected in IgG. A few N-glycans unique to EVs were highly sialylated (i.e., 5−14 SiA residues) N-glycans. A thorough quantitative differential analysis of glycan abundances was also performed (see Supplementary Fig. 1B for the relative standard deviations (RSDs) of glycan abundances measured in the CE-MS analyses of the blood isolates). Similarities were observed in the relative abundance levels of the N-glycans detected in the four analyzed blood-derived samples (Supplementary Data 2). Several glycans, including FA2G2S2, FA2BG2S2, A2G2S2, and A3G3S3, were detected at high abundance,

and other glycans, including A4G4S4, FA4G4S4, A3G3S2, and FA3G3S2, were detected at medium abundance in each analyzed isolate sample. Other glycans, e.g., the set of fucosylated analogs of A3G3S3, exhibited different relative abundances in the four examined blood isolate types. As an illustration, FA3G3S3 was detected at lower abundance in the IgM and IgG isolates, compared to plasma and EVs. F2A3G3S3 was not detected in the IgG isolate and detected at relatively low abundance in the other three blood isolates. F3A3G3S3 was detected only in plasma and EV isolates at relatively low abundance, and F4A3G3S3 was detected only in total plasma. Several neutral glycans also exhibited significantly different abundance levels in the four types of analyzed blood-derived isolates. For instance, Man10 was detected at much lower abundance in the IgM and EV isolates, compared to the IgG isolate and total plasma. Man9 was detected at higher abundance in the IgG isolate, compared to the other three blood isolates, while Man8 and Man7 were detected only in the IgG isolate (Supplementary Data 2). Overall, the above-described results demonstrate that the developed workflow is a powerful, straightforward, flexible, and highly sensitive approach to decipher the N-glycomes of complex biological samples. It allowed us to further expand the glycan coverage of the four analyzed blood-derived isolates, including human plasma-derived EVs, using minute amounts of samples, and identify glycan species unique to a specific type of a blood isolate.

## Single mammalian cell N-glycome profiling

**Single-cell loading and in-capillary N-glycan release.** Individual mammalian cells were introduced into the CE capillary in a controlled manner using a hydrodynamic injection mode (Fig. 1A and Supplementary Fig. 2A, B), as described in the Methods section. To visually confirm the cell loading process, the cells were treated with a plasma membrane-binding dye and subsequently imaged using bright-field and fluorescence microscopy techniques (Fig. 1C, D). Upon introduction into the capillary, the evaluated cells exhibited a tendency to weakly and transiently adhere to the capillary surface if the hydrodynamic flow was either halted or maintained at an excessively low rate. This phenomenon of cell immobilization is likely attributable to the formation of hydrogen bonds and van der Waals interactions between the silanol groups of the bare fused silica capillary surface and numerous chemical groups present on the cell surface. According to our measurements made under the described experimental conditions, larger HeLa and U87 cells may possess a diameter ≥24.0 μm and ≥29.0 μm, respectively (see "Methods" section). When the diameter of the injected cells was very close to or larger than the internal diameter of the capillary (i.e., 30 μm), cells got slightly squeezed upon entering the capillary, and their size could further impede their mobility. These tendencies (cell adherence to the capillary wall and cell squeezing during the cell loading process for larger cells) were strategically exploited to facilitate the cell stacking for the effective injection of several cells and to better control the distance between the injected individual cells and the capillary inlet, thus making each injection more reproducible. No noticeable differences in stacking (or adherence to the capillary wall) of smaller and larger cell populations were observed in our proof-of-concept experiments conducted using HeLa and U87 cell lines.

The cells loaded into the CE capillary were sandwiched between two plugs of a PNGase F digestion solution, and two short CE voltage pulses (30 s each) were applied in normal and reverse polarity to effectively mix the cells with the glycosidase (see "Methods" section). No lysis buffer was employed and/or injected to preserve the cell integrity and release only the N-glycans from the cell surface. Ideally, to preserve cellular integrity, the cells should be maintained in a buffer solution that closely mimics physiological pH and osmolarity. However, such buffers are typically incompatible with MS or CE analysis and may result in ionization suppression, adduct formation, and decreased separation performance phenomena. In this study, a stacking strategy was used to increase the peak intensities and

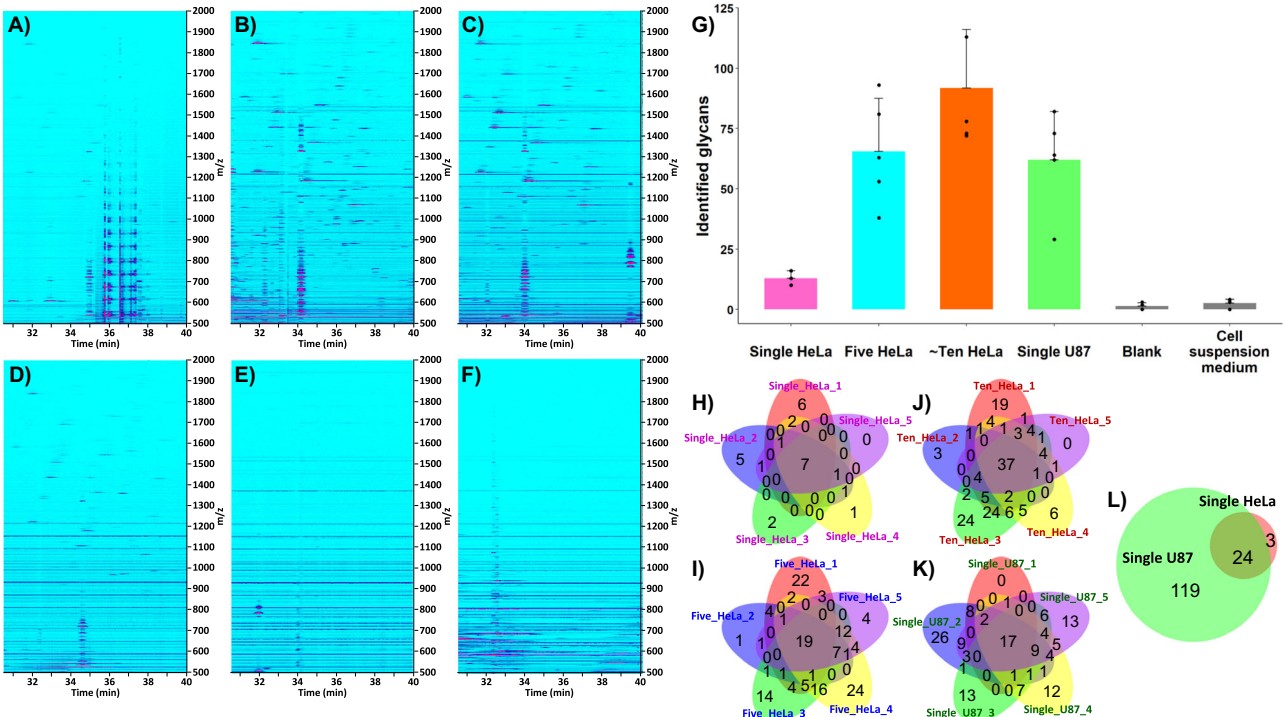

**Fig. 3 | CE-MS-based N-glycan profiling of HeLa and U87 mammalian cells.**
**A–F** Representative ion density maps acquired in label-free CE-MS analyses of N-glycans released from **A** one HeLa cell, **B** five HeLa cells, **C** ten HeLa cells, **D** one U87 cell, **E** water blank sample, and **F** cell suspension medium (a normalized intensity level of $3.9 \times 10^4$ was selected for each density map). **G** Number of N-glycans identified in the analyzed mammalian cell and control samples ($n = 5$ technical replicates, data are presented as mean values ± SD, black dots correspond to individual data points). **H–K** Venn diagrams illustrating the overlap of N-glycans identified in five repetitive analyses acquired with the injection of one (**H**), five (**I**), and -ten (**J**) HeLa cells, and one U87 cell (**K**). **L** Venn diagram illustrating the overlap of N-glycans identified in single HeLa and single U87 cells, based on the total number of N-glycans identified in HeLa and U87 single cells (see **H** and **K**, respectively). Source data are provided as a Source Data file.

enhance the peak shape for optimized detection and separation of the released glycans. To enable this strategy, the cells were resuspended in 1 mM ammonium acetate pH 6.7, immediately prior to their loading into the CE capillary, and the commercial PNGase F enzyme was diluted 7-fold in water to highly decrease the salt concentration (see "Methods" section). Given that the cells were exposed to a low osmolarity environment during the deglycosylation step for N-glycan release, an assessment of the post-incubation cell integrity was conducted through the offline incubation of single cells for 1 h, using the conditions employed in the in-capillary sample processing workflow (i.e., the cells were sandwiched between two PNGase F plugs). Fluorescence imaging of the single cells prior to and after offline incubation in a small piece of capillary did not reveal discernible alterations in the cell morphology, size, or membrane integrity (Fig. 1D). To further check the cell viability, morphology, and membrane integrity under the selected in-capillary sample processing conditions, a suspension of HeLa cells stained with a fixable dead cell dye (see "Methods" section) was incubated for 1 h with PNGase F in 1 mM ammonium acetate pH 6.7. Based on microscopy visualization (Supplementary Fig. 2E, F), we estimated that >40% of the HeLa cells were still alive after the deglycosylation step with PNGase F. We also noticed that the majority of the dead cells exhibited a morphology very similar to that of the live cells, suggesting that the cell integrity could be preserved under our experimental conditions for N-glycan release. Therefore, we concluded that the majority of detected and identified glycans were removed from the cell surface in our in-capillary cell processing proof-of-concept experiments.

**N-glycan profiling of single, five, and ~ten HeLa cells.** To assess the capability of the developed workflow for direct and unbiased N-glycan profiling of mammalian cells, sets of five repetitive experiments were performed with the injection of one (Supplementary Fig. 2A), five, and -ten (i.e., $10 \pm 4$ cells, referred to as "bulk sample," see "Methods" section) HeLa cells. Characteristic ion density maps acquired in CE-MS analysis of HeLa cell-derived N-glycans are presented in Fig. 3A–C (see Supplementary Note 2). Processing of CE-MS[1] data resulted in the identification of $13 \pm 3$ (one HeLa cell), $66 \pm 22$ (five HeLa cells), and $92 \pm 24$ (-ten HeLa cells) N-glycans ($n = 5$), respectively (Fig. 3G and Supplementary Data 1). As expected, due to cell-to-cell heterogeneity (arising from, inter alia, molecular variability, and cell-cycle position) and cell size variations (the surface areas of the injected HeLa cells being in the range 1282–3258 μm², Supplementary Fig. 2C), the number and type (i.e., monosaccharide composition) of identified N-glycans exhibited significant variations. Figure 3H–J depicts the overlap of the N-glycans detected in the five repetitive analyses acquired for each HeLa cell loading. In total, 27, 148, and 160 non-redundant N-glycan compositions were identified in one, five, and -ten HeLa cells, respectively. As shown in Fig. 3E, G and Supplementary Data 1, the levels of carryover derived from the analysis of preceding HeLa cell samples were insignificant (or below the limit of detection (LOD) of the applied CE-MS method) while using the described capillary rinse cycles, based on control analyses performed with a water blank sample (see "Methods" section). CE-MS analyses of the cell suspension medium, collected from the same cell suspension used to inject individual cells, were also carried out. These experiments allowed us to confirm that the detected glycans were not derived from extracellular proteins or contaminants present in the cell medium (Fig. 3F, G, and Supplementary Data 1).

Higher levels of fucosylation (up to 6 fucose residues) and sialylation (5–12 SiA residues) were detected in 5–10 HeLa cells, compared to single HeLa cells, for which the degrees of fucosylation and sialylation of identified N-glycans did not exceed 2 and 4, respectively

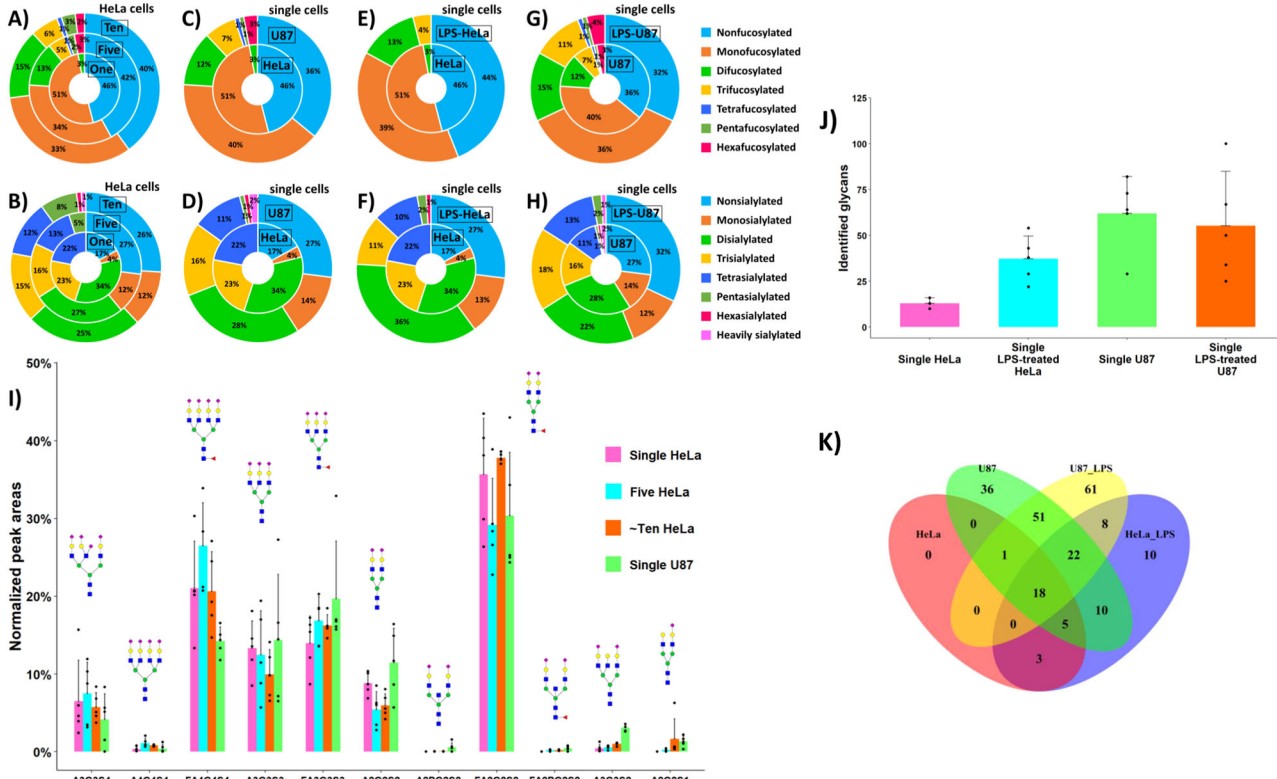

**Fig. 4 | Fractional distributions of fucosylated and sialylated N-glycans detected in HeLa and U87 mammalian cells. A–G** Fractional distributions of fucosylated N-glycans detected in one, five, and -ten HeLa cells (**A**), HeLa and U87 single cells (**C**), LPS-treated and untreated single HeLa cells (**E**), and LPS-treated and untreated single U87 cells (**G**). **B–H** Fractional distributions of sialylated N-glycans detected in one, five, and -ten HeLa cells (**B**), HeLa and U87 single cells (**D**), LPS-treated and untreated single HeLa cells (**F**), and LPS-treated and untreated single U87 cells (**H**). **I** Normalized abundances of eleven representative N-glycans detected in one, five, and -ten HeLa cells, and one U87 cell (*n* = 5 technical replicates, data are presented as mean values ± SD, black dots correspond to individual data points). Glycan symbols: blue square, GlcNAc; red triangle, Fuc; green circle, Man; yellow circle, Gal; purple diamond, Neu5Ac. **J** Number of N-glycans identified in single HeLa and single U87 cells before and after LPS stimulation (*n* = 5 technical replicates, data are presented as mean values ± SD, black dots correspond to individual data points). **K** Venn diagram illustrating the overlap of N-glycans identified in single HeLa and single U87 cells with and without LPS treatment, based on the total number of N-glycans identified in the respective analyzed samples. Source data are provided as a Source Data file.

(Fig. 4A, B). These results indicated the extremely low abundance levels of highly fucosylated and highly sialylated N-glycans in HeLa cells since they could not be detected at the single HeLa cell level using our developed workflow. Mono- and difucosylated N-glycans accounted for 51% and 3% of the total glycans detected in single HeLa cells, respectively (46% of glycans were nonfucosylated). For 5 and 10 HeLa cells, the fractional distributions of fucosylated N-glycans were 34%, 13%, 5%, 1%, 2%, and 3% (five HeLa), and 33%, 15%, 6%, 1%, 3%, and 2% (-ten HeLa) for mono-, di-, tri-, tetra-, penta-, and hexafucosylated N-glycans, respectively (Fig. 4A). Mono-, di-, tri-, and tetrasialylated N-glycans accounted for 4%, 34%, 23%, and 22% of the total glycans detected in single HeLa cells, respectively (17% of glycans were non-sialylated). For 5 and 10 HeLa cells, the fractional distributions of sia-lylated N-glycans were 12%, 27%, 16%, 13%, and 5% (five HeLa), and 12%, 25%, 15%, 12%, 8%, 1%, and 1% (-ten HeLa) for mono-, di-, tri-, tetra-, penta-, hexasialylated, and heavily sialylated (≥7 SiA residues) N-glycans, respectively (Fig. 4B).

As expected, due to the cell-to-cell heterogeneity, high variations in the raw (i.e., non-normalized) glycan abundances measured in the five repetitive analyses were observed for one, five, and -ten injected HeLa cells. For instance, the mean RSD of peak areas for eight selected individual representative glycans accounted for 103% in the measurements of single HeLa cells (Supplementary Fig. 3A and Supplementary Note 3). We hypothesize that such significant variation might be mostly attributed to the cell size, surface area, and cell-cycle state. Nevertheless, a

substantial increase in the glycan abundance levels was demonstrated with increased loaded cell numbers. As an illustration, Supplementary Fig. 3C shows the summed raw abundances of the same selected representative N-glycans, detected at high abundance in one, five, and -ten HeLa cells. A linear relationship was demonstrated between the injected cell numbers and the total cellular glycan amounts for the eight selected glycans, based on peak area measurements (Supplementary Fig. 3D and Supplementary Note 3). As shown in Supplementary Fig. 3B, the normalization of glycan abundances resulted in significantly lower RSDs of glycan abundances, in comparison to the RSDs of raw abundances, as illustrated with the eight selected representative glycans. A systematic normalization of glycan abundances was therefore performed for the relative quantitative comparison of the N-glycan profiles detected in single cells and 5–10 cells. Figure 4I displays the normalized abundances of eleven representative N-glycans detected in one, five, and -ten HeLa cells (i.e., each glycan abundance was normalized with respect to the summed abundances of the eleven selected glycans). FA2G2S2, A3G3S3, FA3G3S3, and FA4G4S4 were detected at high abundance levels in the analyzed HeLa cells. To the contrary, A2G2S1, A2BG2S2, FA2BG2S2, A3G3S2, and A4G4S4 were measured at low abundances. Overall, the relative abundances of the eleven selected HeLa cell-derived N-glycans were consistent across different cell loading levels, based on peak area measurements. For instance, the relative abundance of FA4G4S4 was approximately

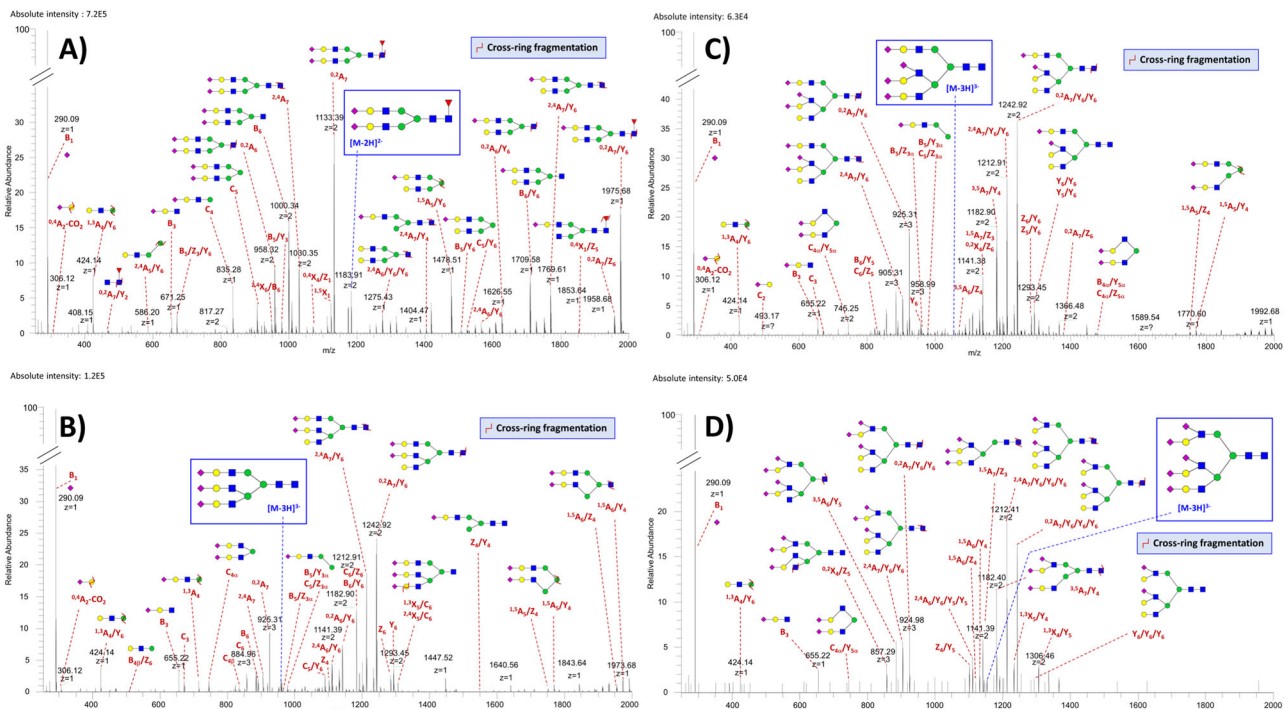

**Fig. 5 | CE-MS²-based structural characterization of representative HeLa and U87 cell-derived N-glycans. A–D** Characteristic CE-MS² spectra of **A** the fucosylated disialylated glycan FA2G2S2 (Fuc₁Hex₅HexNAc₄Neu5Ac₂), **B** the trisialylated glycan A3G3S3 (Hex₆HexNAc₅Neu5Ac₃), **C** the tetrasialylated glycan A3G3S4 (Hex₆HexNAc₅Neu5Ac₄), and **D** the pentasialylated glycan A3G3S5 (Hex₆HexNAc₅Neu5Ac₅), selecting the molecular ions at *m/z* 1183.42, 958.66,

1055.69, and 1152.73 as precursor ions, respectively. Fragment ions are annotated based on the Domon and Costello nomenclature. Blue square, GlcNAc; red triangle, Fuc; green circle, Man; yellow circle, Gal; purple diamond, Neu5Ac. Symbol Z indicates cross-ring fragmentation. Only the most intense/relevant fragments are annotated in the shown spectra.

twice lower than that of FA2G2S2 but about three times higher than that of A3G3S4 in one, five, and ~ten HeLa cells.

CE-MS² analyses of HeLa cell-derived N-glycans were performed to confirm the N-glycan composition identification results and provide information on structural features of the detected glycans (e.g., antenna-branching, fucose position, and SiA linkage). Proof-of-concept CE-MS² experiments performed with ~ten HeLa cells resulted in the accurate and unambiguous structural characterization of 53 N-glycans in HeLa cells (~60% of the glycans detected and identified in CE-MS¹ analyses were structurally characterized by CE-MS²), including negatively-charged (i.e., sialylated) and neutral glycans (Supplementary Data 3 and Supplementary Note 4). As expected, neutral glycans (mobilized under the applied electric field through ion-dipole interactions with acetate anions present in the BGE[40]) migrated later than sialylated glycans. Figure 5 shows characteristic MS² spectra of three representative sialylated N-glycans detected in HeLa cells. The molecular ions at *m/z* 1183.42, 958.66, and 1055.69 were selected as precursor ions for the MS²-based structural characterization of FA2G2S2 (Mr_th 2368.84 Da, Fig. 5A), A3G3S3 (Mr_th 2879.01 Da, Fig. 5B), and A3G3S4 (Mr_th 3170.11 Da, Fig. 5C), respectively. The MS² spectra of these sialylated glycans all exhibited a predominant fragment ion B₁¹⁻ at *m/z* 290.09, corresponding to the loss of one SiA residue at the termini of the antennae, and the diagnostic ion ⁰,⁴A₂-CO₂¹⁻ at *m/z* 306.12, revealing the presence of α-2,6 5-N-acetyl-neuraminic acid (Neu5Ac) linkages[55]. In the mass spectra of the fucosylated FA2G2S2 glycan, the characteristic mass difference of 206.08 Da between ²,⁴A₇²⁻ (*m/z* 1029.85) and ⁰,²A₇²⁻ (*m/z* 1132.89), and ²,⁴A₇/Y₆¹⁻ (*m/z* 1769.61) and ⁰,²A₇/Y₆¹⁻ (*m/z* 1975.68) ion pairs located the fucose on the chitobiose core[40,56]. For the tri-antennary A3G3S3 glycan, the diagnostic ions B₅/Z₃α¹⁻ (*m/z* 961.31), B₅/Y₃α¹⁻ (*m/z* 979.32), and C₄α²⁻ (*m/z* 745.25) allowed us to assign a branched 3-linked antenna[40,56,57]. Similarly, the fragment ions B₅/Z₃α¹⁻ (*m/z* 961.31), B₅/Y₃α¹⁻ (*m/z* 979.32), and C₄α/Y₅α²⁻

(*m/z* 745.25) were diagnostic of a branched 3-linked antenna for the tri-antennary A3G3S4 glycan. The pentasialylated glycan A3G3S5, a complex glycan that was rarely structurally characterized by tandem MS in the past[40], was also detected in HeLa cells (see below and Fig. 5D for the MS² fragmentation pattern of this glycan). CE-MS² data also allowed us to structurally characterize neutral N-glycans in HeLa cells, including high-mannose-type N-glycans, from Man2 to Man13, with or without a bisecting GlcNAc residue (Supplementary Data 3). Supplementary Fig. 4 depicts the fragmentation patterns of Man6, Man7, and Man13. For these glycans, characteristic fragment ions (e.g., at *m/z* 179.06, 323.10, 485.15, 869.27, and 1031.33) were supportive of a branched oligomannosyl structure[58]. Finally, ~70% of the structures characterized by tandem MS in HeLa cells were complex-type glycans, ~21% high-mannose-type glycans, and ~9% hybrid-type glycans. These results further demonstrate the detection capabilities of the developed technique for diverse glycan types, with significantly different monosaccharidic compositions and structures.

**Differential N-glycome analysis of single HeLa and U87 cells.** Next, the developed in-capillary workflow was applied to the CE-MS analysis of single U87 cells to assess, as a proof-of-concept, whether we could detect qualitative and/or quantitative differences in cell surface N-glycomes of different cell types at the single-cell level. In comparison to single HeLa cells, a significantly higher number (~5-fold) of N-glycans were detected and identified in single U87 cells. The five repetitive experiments carried out with single U87 cells (Supplementary Fig. 2B) resulted in the detection of 62 ± 20 N-glycans per single-cell (Fig. 3D, G and Supplementary Data 1). In total, 143 non-redundant N-glycan compositions were identified in the analyzed single U87 cells (Fig. 3K). The examination of HeLa and U87 cell-containing droplets under a bright-field microscope showed that the two cell lines exhibited similar morphological characteristics

in suspension but distinct size distributions. The average diameters and surface areas of representative HeLa and U87 single cells were determined to be $21.8 \pm 4.9 \, \mu m$ and $1563 \pm 763 \, \mu m^2$ (HeLa), and $26.1 \pm 6.6 \, \mu m$ and $2282 \pm 1266 \, \mu m^2$ (U87), respectively (see "Methods" section and Supplementary Fig. 2C, D). The higher number of N-glycans detected in single U87 cells could, therefore, be attributed in part to the larger size of this cell type (~2-fold higher surface area), compared to HeLa cells. Yet, the non-linear relationship between the number of detected glycans and the size of the analyzed mammalian cells indicated that, as expected, other factors, e.g., unique molecular features and their abundances specific to HeLa and U87 cells, contributed to the significantly different numbers of N-glycans detected at the surface of HeLa and U87 single cells. Besides, the intrinsic cell morphology and structural characteristics of each cell type might also create locus-dependent steric hindrances at the surface of the single cells, resulting in differential accessibility of PNGase F to these specific cell surface loci. We expect to observe the increased depth of N-glycome profiling at the levels of five and ten U87 cells (and potentially other cell lines), similarly to the determined trends in the discussed above HeLa experiments. However, examining the glycan profiling trends for profiling of single and multiple cells experimentally was not the focus of this proof-of-concept analysis of U87 cells.

89% of the glycans identified in single HeLa cells were also detected in single U87 cells (Fig. 3L). These commonly detected glycans accounted for 17% of the total number of N-glycans identified in single U87 cells, indicating that 83% of the glycans detected in U87 cells were specific to this cell line or could not be detected in HeLa cells due to their very low abundance or under-representation in single HeLa cells. In single U87 cells, higher levels of fucosylation (up to 6 fucose residues) and sialylation (up to 12 SiA residues) were detected, compared to single HeLa cells (Fig. 4C, D). Consequently, the unique glycans detected in single U87 cells encompassed tri-, tetra-, penta-, and hexafucosylated glycans, and sialylated glycans composed of 5-12 SiA residues, which were not detected in single HeLa cells. In single U87 cells, mono-, di-, tri-, tetra-, penta-, and hexafucosylated N-glycans accounted for 40%, 12%, 7%, 1%, 1%, and 3%, respectively (36% of glycans were nonfucosylated, Fig. 4C). The fractional distributions of sialylated N-glycans were as follows: 14% (mono-), 28% (di-), 16% (tri-), 11% (tetra-), 1% (penta-), 1% (hexa-), and 2% (heavily sialylated, i.e., ≥7 SiA residues), respectively (27% of glycans were nonsialylated) (Fig. 4D).

Noticeable differences in the abundances of the N-glycans detected in HeLa and U87 single cells were also observed, based on peak area measurements. As an illustration, Fig. 4I shows the relative abundances of eleven selected N-glycans commonly detected in HeLa and U87 single cells. Based on the statistical paired t-test (Supplementary Data 4A, B), a couple of glycans were detected in significantly higher or noticeably higher abundance levels in U87 cells, e.g., FA2BG2S2, A3G3S2, and A2G2S1 glycans ($p < 0.05$), and FA3G3S3 and A2BG2S2 ($p = 0.1$). On the contrary, the tetrasialylated glycan FA4G4S4 exhibited a higher abundance in HeLa cells ($p = 0.06$).

Figure 6A depicts the results of non-supervised Euclidean distance-based hierarchical clustering of quantitative glycomic profiles of 47 representative N-glycans detected in HeLa and U87 single cells (i.e., glycans that were detected in at least three CE-MS analyses out of the ten total repetitive analyses). This clustering analysis yielded a reasonable differentiation of HeLa and U87 cell lines, according to their respective single-cell N-glycome profiles. As shown in Fig. 6A and Supplementary Data 5A, HeLa and U87 single cells were clustered into two distinct clades, based on five repetitive analyses acquired for each single-cell type. Glycans like FA2G2S2, FA3G3S3, and FA4G4S4 were clustered into the same clade due to relatively high-intensity levels in both HeLa and U87 cell types. Other clades reflected similar intensity levels of relatively medium abundance glycans, e.g., FA3G3S2 and FA4G4S3, in the two analyzed cell types. Interestingly, Man3 and the fucosylated analogs of Man6 and Man12 were only detected in single

U87 cells and not in single HeLa cells, while Man6, Man8, and FMan7 were detected in both cell types at similar intensity levels but with a higher frequency in single U87 cells. The previously reported studies on the glycomic analysis of U87 or other cells focused on specific proteins' glycosylation and its biological significance did not provide information on cell surface glycans' abundances[59–61]. In contrast, our results present an inventory of single HeLa and single U87 cell surface glycans with their respective glycan compositions and abundances, which allows us to highlight significant and potentially biologically relevant glycosylation differences between the two cell lines. Principal component analysis (PCA) was also conducted to visualize the dominant trends and underlying specific patterns in the datasets generated by HeLa and U87 single-cell analyses (Fig. 6B). Selecting the 47 representative N-glycans detected in HeLa and U87 single cells above-described (Fig. 6A), two distinct clusters could be clearly differentiated with PCA, corresponding to HeLa and U87 cell lines, respectively (Fig. 6B). The PCA algorithm, therefore, allowed us to confirm the uniqueness and specificity of HeLa and U87 single-cell N-glycomes.

Finally, our proof-of-concept CE-MS² experiments were performed with ~ten U87 cells and resulted in accurate and unambiguous structural characterization of 29 N-glycans, including sialylated and neutral N-glycans (Supplementary Data 6). Figure 5D shows a characteristic MS² spectrum of A3G3S5, detected at a relatively medium abundance in U87 cells. The MS²-based structural characterization of this pentasialylated glycan was based on the $[M-3H]^{3-}$ precursor ion at $m/z$ 1152.73. For this glycan, the detection of the $C_{4\alpha}/Y_{5\alpha}^{2-}$ fragment ion at $m/z$ 745.25 and the absence of $C_{4\beta}/Y_{5\beta}^{1-}$ ion at $m/z$ 835.28 allowed the assignment of a branched 3-linked antenna[40]. The typical mass difference of 60.02 Da between the pair of ions $^{2,4}A_7/Y_6/Y_6^{3-}$ ($m/z$ 904.97) and $^{0,2}A_7/Y_6/Y_6^{3-}$ ($m/z$ 924.98), and $^{2,4}A_7/Y_6/Y_6/Y_6^{2-}$ ($m/z$ 1212.41) and $^{0,2}A_7/Y_6/Y_6/Y_6^{2-}$ ($m/z$ 1242.42) confirmed the absence of a core fucose[56]. Interestingly, $^{0,4}A_2$-$CO_2^{1-}$ diagnostic ion at $m/z$ 306.12 was missing despite the relatively high intensity of $B_1^{1-}$ ion at $m/z$ 290.09 (i.e., compared to the intensity level of $B_1^{1-}$ ion detected in A3G3S4 fragmentation spectrum (Fig. 5C), in which $^{0,4}A_2$-$CO_2^{1-}$ ion could be detected). These results revealed the presence of α-2,3 SiA linkages in A3G3S5.

## Single-cell N-glycome alterations induced by LPS stimulation.

So far, to the best of our knowledge, there are no clear-cut benchmarking methods that involve precise manipulation of induced glycome changes (e.g., genetic perturbation methods or biochemical/biophysical techniques) leading to predictable, highly similar, and reproducible glycan profile alterations across various cell types that can be used at a single-cell level analysis. Previous studies reported that THP-1 mammalian cells treated with lipopolysaccharide (LPS) exhibited increased[62] or decreased[63] levels of sialylation. Downregulation of glycan fucosylation was also reported for LPS-stimulated brain cells[64]. To check if the developed CE-MS-based workflow could detect glycome alterations at the single-cell level, HeLa and U87 cells were stimulated with LPS. Interestingly, N-glycan profiling of single HeLa cells, after stimulation of HeLa cells with LPS, resulted in an ~3-fold increased number of detected N-glycans, compared to the untreated HeLa cells (Fig. 4J and Supplementary Data 1). On average, $37 \pm 12$ N-glycans per single HeLa cell ($n = 5$) were identified after LPS treatment, and 76 non-redundant N-glycan compositions were identified in total (Fig. 4K). As shown in Fig. 4E, mono-, di-, and trifucosylated glycans were detected in single HeLa cells following LPS treatment, while N-glycan profiling of single HeLa cells before LPS treatment resulted only in the assignment of mono- and difucosylated glycans. LPS activation of HeLa cells also resulted in the detection of penta- and hexasialylated glycans, undetectable in single HeLa cells before LPS treatment (Fig. 4F). Overall, the fractional distributions of N-glycans detected in single HeLa cells were significantly different before and after LPS treatment. Mono-, di-, and trifucosylated N-glycans accounted for 39%, 13%, and 4%, respectively, in LPS-treated HeLa cells (vs. 51%, 3%, and 0%, respectively, in

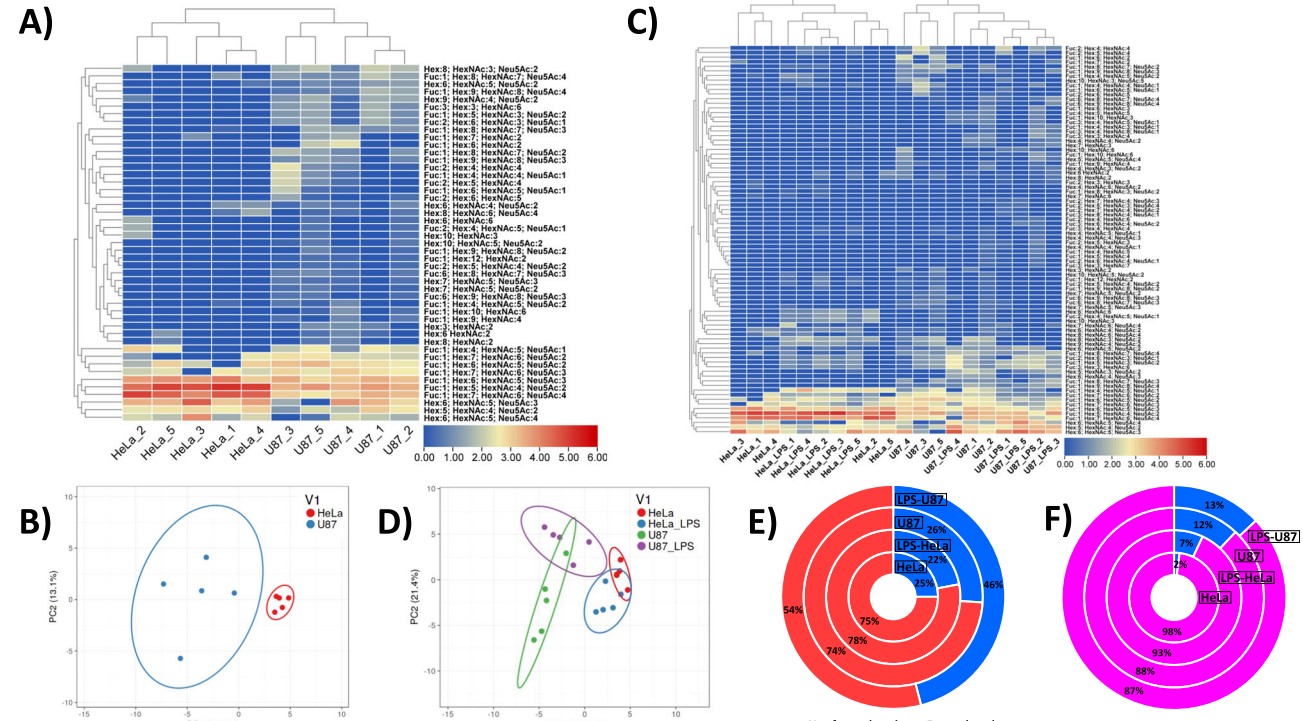

**Fig. 6 | Differential qualitative and quantitative N-glycan profiling of single HeLa and single U87 mammalian cells. A–C** Euclidean distance-based hierarchical clustering of N-glycans detected in **A** HeLa and U87 cells, and **B** LPS-treated and untreated HeLa and U87 cells, based on five repetitive analyses of single HeLa and single U87 cells (labeled from 1 to 5). Red, yellow, and light blue colors (corresponding to log$_2$ values of the normalized signal intensities ranging from 6 to 3, 3 to 2, and 2 to 0.04, respectively) display high, medium, and low relative abundances, based on the normalized N-glycan signal intensities (see "Methods" section and Supplementary Data 5). N-glycans that are not detected in the cell samples are highlighted in dark blue. **B–D** Principal component analysis (PCA) of N-glycans detected in HeLa and U87 cells (**B**), and PCA of LPS-treated and untreated HeLa and U87 cells (**D**). **E, F** Total abundances of fucosylated and nonfucosylated (**E**) and sialylated and nonsialylated (**F**) N-glycans detected in HeLa and U87 single cells, before and after LPS treatment, based on the normalized N-glycan signal intensities.

untreated HeLa cells, Fig. 4E). Mono-, di-, tri-, tetra-, penta-, and hexa-asialylated N-glycans accounted for 13%, 36%, 11%, 10%, 2%, and 1%, respectively, in LPS-treated HeLa cells (vs. 4%, 34%, 23%, 22%, 0%, and 0%, respectively, in untreated HeLa cells, Fig. 4F). As shown in Supplementary Fig. 5A, 34% of the N-glycans detected in LPS-treated HeLa cells were also identified in untreated HeLa cells. Interestingly, 50 N-glycans were uniquely detected in LPS-treated HeLa cells, among which 44% were neutral (including trifucosylated) glycans, and 56% were sialylated (including penta- and hexasialylated) glycans. A quantitative comparison of glycan abundances was also conducted. As shown in Fig. 6E and Supplementary Fig. 5C, the total abundances of fucosylated glycans detected in single HeLa cells were not significantly different ($p = 0.6$, Supplementary Data 4C, D) before and after LPS treatment, and accounted for 75% and 78%, respectively, based on peak intensity measurements. However, the stimulation of HeLa cells with LPS resulted in significantly altered sialylation profiles. As shown in Fig. 6F, a high increase in the total abundance of HeLa cell-derived nonsialylated glycans was noticed ($p = 0.06$, Supplementary Data 4F, G) after LPS treatment. In addition, the total abundances of mono-, tri-, and tetrasialylated glycans were significantly different ($p < 0.05$, Supplementary Data 4F, G) in untreated and LPS-treated HeLa cells (Supplementary Fig. 5D). These results confirmed that LPS stimulation of HeLa cells induced significant changes in HeLa cell N-glycome profiles, which could be detected at the single-cell level using our proof-of-concept workflow.

Significant alterations of U87 cell N-glycome profiles were also observed at the single-cell level when U87 cells were treated with LPS, compared to the untreated U87 cells. CE-MS analysis of single U87 cells after LPS treatment resulted in the detection of 55 ± 30 N-glycans per single U87 cell ($n = 5$), and in the assignment of 161 non-redundant N-glycan compositions in total (Fig. 4J, K and Supplementary Data 1). 68% of the N-glycans identified in LPS-treated U87 cells were fucosylated, and the fractional distributions were 36%, 15%, 11%, 1%, 1%, and 4% for mono-, di-, tri-, tetra-, penta-, and hexafucosylated N-glycans, respectively (Fig. 4G). Mono-, di-, tri-, tetra-, penta-, hexa-, and heavily (i.e., ≥7 SiA residues) sialylated N-glycans accounted for 12%, 22%, 18%, 13%, 2%, 0%, and 1%, respectively (Fig. 4H). As shown in Supplementary Fig. 5B, 92 glycans were commonly detected in LPS-treated and untreated U87 cells. Yet, 69 glycans were uniquely detected in LPS-treated U87 cells, among which many pentasialylated and several heavily sialylated N-glycans containing up to 13 SiA residues (interestingly, ~90% of these unique glycans were not detected either in LPS-treated HeLa cells, Fig. 4K). These results seem to indicate that LPS treatment induced a modification in the type and structure of biosynthesized sialylated glycans in U87 cells. We think that these unique highly/heavily sialylated glycans, detected on the surface of LPS-stimulated U87 cells at the single-cell level using our developed method, might be challenging to detect using alternative total cellular glycomic analysis or lectin-based methodologies. Indeed, based on our results, the fractional distributions (Fig. 4G, H) and fractional abundances (Fig. 6F and Supplementary Fig. 5F) of sialylated N-glycans identified in single U87 cells did not change significantly before and after LPS treatment (Supplementary Data 4F, H), and only a thorough qualitative and quantitative comparison of released single-cell surface N-glycans could reveal such sialylation subtlety. In addition, the comparative quantitative analysis of fucosylated glycans detected in single U87 cells showed that their total abundances significantly decreased from 74% to 54% ($p < 0.05$, Supplementary Data 4C, E) when U87 cells were treated with LPS,

based on peak intensity measurements (Fig. 6E and Supplementary Fig. 5E). These results clearly demonstrated that LPS treatment induced a downregulation of fucosylated glycans in U87 cells that could be detected at the single-cell level.

Figure 6C and Supplementary Data 5B depict the results of Euclidean distance-based hierarchical clustering of quantitative glycomic profiles of 84 representative N-glycans detected in LPS-treated and untreated HeLa and U87 single cells (i.e., the set of 47 glycans highly representative of HeLa and U87 cells, described above, was extended with 37 representative glycans detected in LPS-treated HeLa and LPS-treated U87 cells, respectively, see "Methods" section). This clustering analysis resulted in a plausible differentiation of two clades, corresponding to treated/untreated HeLa cells, and treated/untreated U87 cells, respectively. In addition, four repetitive analyses of LPS-treated HeLa cells and four repetitive analyses of LPS-treated U87 cells were clustered together. These results further demonstrated the uniqueness of HeLa and U87 cell line N-glycomes and showed that both cell lines exhibited different biological responses to LPS treatment. PCA was also conducted on the datasets generated from the analyses of LPS-treated and untreated HeLa and U87 cells, selecting 57 representative glycans (see "Methods" section) (Fig. 6D). As shown in Fig. 6D, four distinct clusters corresponding to LPS-treated and untreated HeLa cells, and LPS-treated and untreated U87 cells, respectively, were differentiated and confirmed a pronounced effect of LPS treatment. As expected, a partial overlap was observed between the two single HeLa cell-related clusters and the two single U87 cell-related clusters. Overall, these results indicated the undeniable biological effect of LPS stimulation on HeLa and U87 cell N-glycomes, which could be detected at the single-cell level using our developed SCG workflow.

## Discussion

Single-cell omics, spatial omics, and multi-omics are emerging fields in life science, medicine, and fundamental biology applications. Deciphering cell-to-cell variations can be crucial in designing advanced approaches for the diagnosis and treatment of human pathologies. Given the relevance of cell glycosylation analysis for the understanding of tumor initiation, progression, and metastasis, and for identifying effective therapeutic targets, there is a crucial need for the development of SCG methods. In this work, we developed an in-capillary sample processing method for straightforward, unbiased, accurate, and deep qualitative and quantitative N-glycan profiling of single mammalian cells with label-free high-sensitivity CE-MS. Native N-glycans were enzymatically released from the cell surface prior to their CE-MS analysis in the described set of proof-of-concept experiments. To the best of our knowledge, analytical technologies enabling direct and unbiased profiling of single-cell N-glycome have not been reported yet. To date, only a few analytical technologies based on carbohydrate-binding lectins have been developed for SCG. These technologies, which require sophisticated instrumentation, are tedious, expensive, and time-consuming and result in undirect and biased profiling of cell surface glycans. Indeed, the glycans are not enzymatically released from the cell surface for their direct detection and quantification, and the glycan detectability is highly dependent on the lectin-binding affinity and the efficiency of binding, which might be influenced by various factors, including the binding site accessibility and steric hindrance. Furthermore, every single glycan structure does not have a corresponding lectin, and many lectins exhibit low binding specificity toward individual oligosaccharides[65]. In its current proof-of-concept implementation, the SCG method we developed requires modest but reasonable time per analysis (~1 h for the CE-MS analysis and ~3 h in total, including cell loading, in-capillary glycan release, and capillary rinses and conditioning), allows a well-controlled injection and processing of individual cells, and requires affordable analytical instruments, compared to the methods reported for total cellular glycomics and lectin-based SCG. Our results showed that the developed CE-MS-based workflow could result in the detection, identification, and quantitation of up to 100 N-glycans in one single mammalian cell using the MS equipment we currently have access to. Owing to the current lack of analytical technologies enabling direct and unbiased SCG profiling, a direct comparison of the numbers of glycans detected in single mammalian cells, using our developed CE-MS method, with published data remains challenging. Nevertheless, it is worth noting that recent studies focused on SCG with lectin-based techniques reported less than 40 lectin-binding signals per single-cell[37]. We would like to emphasize that this proof-of-concept study explores uncharted territories of MS-based SCG profiling, where the intercellular alterations of glycomes are hitherto unknown. The RSDs of glycan abundances measured in our proof-of-concept SCG experiments are similar to the reported single-cell proteomic abundance measurements[66–68] and substantially more accurate than the RSD values of quantitative measurements observed and reported in recent single-cell transcriptomic studies, which might be well over 600%[19,69–71]. Expectedly, larger-scale studies involving a larger variety of cell lines and higher numbers of experimental repetitive CE-MS analyses of single cells (or tiny amounts of cells) will be required in the future to more accurately determine the RSD values of quantitative measurements using our developed SCG profiling technique. Future investigations will allow us to assess the scalability and throughput of our developed CE-MS-based technology. We expect that the N-glycan profiling of small cell populations of ~50 cells (or more) can be achieved using our developed workflow and automated cell injections. We also believe that the developed workflow could be applied to the analysis of single cells in samples containing heterogeneous cell populations, including tissue samples dissociated into cell suspensions.

Specific N-glycosylation patterns were demonstrated for HeLa and U87 single cells, based on a thorough differential analysis of qualitative and quantitative single-cell N-glycome profiles. A substantially higher number of N-glycans (~5-fold) was detected on the surface of single U87 cells, compared to single HeLa cells, which may be attributed in part to the larger size of U87 cells. In addition, significant differences in the fractional distributions and abundances of the N-glycans detected in HeLa and U87 single cells were observed, reflecting unique molecular features for each cell type. Interestingly, N-glycome alterations were observed at the single-cell level when HeLa and U87 cells were stimulated with LPS, which manifested the change in the phenotypic cell state reflected on the cell surface. Notably, a significantly higher (~3-fold) number of N-glycans and higher levels of fucosylation and sialylation were detected in LPS-treated HeLa cells, compared to untreated HeLa cells. On the other hand, the stimulation of U87 cells with LPS induced the downregulation of fucosylated glycans, compared to the untreated U87 cells. Overall, we demonstrated in the presented here proof-of-concept study that our developed SCG workflow could effectively and accurately characterize the single-cell N-glycome of different mammalian cell lines and detect N-glycome alterations at the single-cell level. The acquired results demonstrated the potential of the technique to differentiate cell phenotypes, states, types, and lineages based on alterations of N-glycan representation on the cell surface. The developed approach is mild, which allowed us to preserve the cell integrity during the enzymatic release of glycans, which may potentially benefit the multi-omic characterization of individual cells (including the combined glycome and proteome analysis of the same single-cell in one single CE-MS analysis, as well as other 'omes'). Future investigations will aim at benchmarking the deglycosylation efficiency of alternative endoglycosidases (e.g., new generations of recombinant PNGase H+ enzymes) to release intact native N-glycans (and potentially O-glycans) from the cell surface, using our developed SCG workflow, and get complementary information on the cell surfaceome (specific cell surface loci accessibility may indeed vary depending on the selected glycosidases).

CE-MS analysis of N-glycans in their native non-labeled state enabled the preservation of glycans' integrity and endogenous structural features, especially fucosylation and sialylation, and allowed us to detect, separate, identify, quantify, and structurally characterize negatively-charged as well as neutral glycans, with a large variety of glycan structural features, from simple bi-antennary structures to highly/heavily branched structures. Highly fucosylated (up to 6 fucose residues) and heavily sialylated (up to 13 Neu5Ac residues) N-glycans were detected in single U87 cells. Highly fucosylated and heavily sialylated N-glycans were also detected in HeLa cells but in larger injected amounts of HeLa cells (5–10 cells), indicating the extremely low abundances of such glycans in HeLa cells. CE-MS$^2$ analyses performed in negative ESI mode resulted in the unambiguous and accurate structural characterization of >60 N-glycans detected in the analyzed mammalian cells and enabled unequivocal identification of core fucose or outer-arm fucose residues and antennary branching. Besides, the presence of α-2,6 Neu5Ac linkages could also be detected in many glycans, based on the detection of the characteristic diagnostic ion $^{0,4}A_2$-$CO_2^{1-}$. CE-MS$^2$ analyses also resulted in the structural characterization of neutral N-glycans, including high-mannose-type N-glycans. We think that with the development of 1- newer generations of mass spectrometers with improved sensitivity and duty cycle; 2- smart on-the-fly MS data acquisition technologies to target glycans for subsequent fragmentation using alternative techniques (e.g., ultraviolet photodissociation (UVPD), electron capture/transfer dissociation (ECD/ETD), etc.); 3- powerful and automated glycan-dedicated software capable of accurately and unambiguously assigning fragments derived from multiple internal fragmentation cleavages; and 4- expanded glycan databases and spectral libraries, encompassing larger varieties of highly/heavily branched and/or highly/heavily sialylated/fucosylated glycan structures, researchers will certainly be able to dramatically increase the numbers of glycan structures fully characterized using the described here CE-MS$^2$ method to analyze mammalian cells or other highly complex and heterogeneous amount-limited biological samples.

In this study, we also demonstrated the potential of the developed workflow for deep and highly informative N-glycan profiling of tiny amounts of human blood-derived isolates, including extracellular vesicles, which are important intracellular communicators with diagnostic potential. Over 132 and 88 N-glycans were detected in IgM and IgG for injected amounts of 100 pg (i.e., 100 amol) of IgM, and 500 pg (i.e., 3 fmol) of IgG, corresponding to only ~10 pg of N-glycans and ~50–60 pL of human serum. These minute amounts of proteins and glycans are estimated to be equivalent to the protein and glycan content of one single mammalian cell. CE-MS analysis of total plasma resulted in the identification of >234 and >152 N-glycans for injected amounts of approximately 50 pL and 5 pL of human plasma, respectively (i.e., ~150 and ~15 pL of blood, respectively). Over 226 and 127 N-glycans were detected in the total EV isolate, using injected amounts equivalent to ~150 nL and ~3 nL of plasma, respectively. The numbers of N-glycans identified in IgM, IgG, total plasma, and total EV isolate reported here, using sub-0.5 ng-levels of serum proteins and nL/pL-levels of plasma isolates, largely exceed (~7-fold) those reported in other N-glycan profiling studies of similar complexity blood-derived isolates[40,47–51,54]. These results further demonstrate the impressively high sensitivity of the developed glycan profiling method, which allowed us to increase the depth of glycan profiling (i.e., detect higher numbers and varieties of glycans) and, therefore, expand the glycan catalog of the four types of analyzed blood-derived isolates. Finally, we believe that our developed in-capillary label-free CE-MS technique highly minimizes potential sources of analytical bias because 1- any sample processing-related analyte alterations and additional sources of pre-analytical variability in glycan identification and quantitation using amount-limited biological specimens are mostly eliminated; 2- the levels of ESI- or in-source-induced decay of labile monosaccharides

(e.g., sialic acid and fucose residues) are negligible (as also shown in our previous study[40]); and 3- the detectability of the released glycans is not dependent on the binding affinity of native glycans toward specific ligands or proteins such as lectins.

We envision that our approach can open new doors in the field of glycomic profiling of scarce samples and single-cell glycomic research, and it can be extended to the analysis of a large variety of glycans (e.g., O-glycans and lipoglycans) as well as to the glycan profiling of other biological and clinically-relevant amount-limited samples. We expect that the developed workflow will provide a wealth of information on eukaryotic (or prokaryotic) cell and EV glycomes and will enable differential glycomic profiling studies of cell and EV subpopulations. The developed method is a promising approach for identifying new glycan biomarkers in human pathologies using limited amounts of biological materials, e.g., liquid microbiopsies, small populations of cells or EVs, and single cells (and even single organelles). The innovative SCG technique could also be potentially integrated with other omic approaches in a single-cell multi-omics platform. Multi-omic and spatial profiling of individual cells could provide crucial information on biological mechanisms underlying complex diseases, which is unachievable by merging data sets obtained from mono-omics studies of different cells and bulk samples.

Over the past decades, chemical tools have been developed to profile cell surface glycans and get insights into their spatial distribution and dynamic turnover for a better understanding of their specific role in the development and progression of human diseases[72]. For example, the Bertozzi group pioneered chemical reporter strategies, including metabolic labeling of cell surface glycans with azido groups, for visualizing and monitoring glycan dynamic changes in living cellular organisms[65,73]. We think that our developed SCG approach could serve as an alternative and complementary technique for a quick, effective, and sensitive analysis of cell surface glycans in small populations of cells and single cells. Our CE-MS-based label-free strategy, which has the capability to detect a large variety of glycans (including peculiar glycans such as heavily sialylated glycans) in their native state, could help monitor cell surface glycosylation changes that regulate cellular functions during cell growth, differentiation, activation, proliferation, and survival. In the emerging fields of glycomedicine and personalized medicine, we think that the developed technique is a promising approach for analyzing the glycome or glycome subtype (e.g., sialome) alterations induced by the chemical and biological treatment of cells with therapeutic drugs and drug candidates that target the level and profile of cellular glycosylation. In the relatively nascent and highly challenging field of SCG, an arsenal of cutting-edge technologies has to be developed and combined in joint efforts for routine screening of single-cell glycomes. We hope that our proof-of-concept study will help shed light on the complexity of cell surface glycans and their roles in the biology of the cell in health and disease.

## Methods

The described research complies with all relevant ethical regulations. The research experiments involving human subjects were reviewed by the respective authorized Institutional Review Boards (IRB) with approvals IRB#2001P000S91 (BIDMC) and IRB#17-12-14 (NU). Twelve self-declared healthy male volunteer donors of various races and ethnicities of the age 23–67 years old were recruited, and informed consent was obtained for each participant. A compensation of $10 was provided to the volunteer donors. No sex or gender analysis was carried out since the comparative analysis of the selected samples was not the goal of this proof-of-concept study.

### Materials and chemicals

Deionized water, methanol (99.9%, LC/MS Grade), Gibco F12-K medium, Gibco DMEM medium, Gibco 0.25% Trypsin-EDTA, Gibco penicillin-streptomycin (P/S), phosphate-buffered saline (PBS),

CellMask™ plasma membrane stain (green), LIVE/DEAD™ fixable green dead cell stain kit, and lipopolysaccharide (LPS) were obtained from Thermo Fisher Scientific (Waltham, MA). 1 N NaOH, 1 N HCl, 5 N ammonium hydroxide, glacial acetic acid (99.99%), ultra-high purity ammonium acetate (99.999%), trypan blue, and total human serum IgM and human serum IgG isolates (purity ≥95%, based on non-reduced SDS-PAGE and verified by nanoLC-MS/MS of tryptic digests) were purchased from Sigma-Aldrich (St. Louis, MO). PNGase F enzyme was from New England Biolabs (Ipswich, MA). Fetal bovine serum (FBS) was purchased from R&D Systems (Minneapolis, MN). Platelet-free anticoagulated with EDTA pooled total human plasma (from blood donated by self-declared healthy male donors of 23–67 years old) was kindly provided by Prof. Ghiran's laboratory (BIDMC, Boston, MA). No protein depletion or enrichment was done prior to the analysis of total plasma samples in this study. All bare fused silica (BFS) capillaries (91 cm × 30 μm i.d. × 150 μm o.d.) with sheathless CESI-MS emitters in OptiMS™ cartridges were from SCIEX (Redwood City, CA). Aquapel® was purchased at Pittsburgh Glass Works (Pittsburgh, PA).

### Cell culture
HeLa-S3 (catalog number: CCL-2.2) and U87-MG (catalog number: HTB-14) cell lines (called HeLa and U87 cells thereafter) were from ATCC (Manassas, VA). HeLa cells were cultured in suspension at 37 °C in a complete F-12K medium supplemented with 10% FBS, 1% P/S, and 5% $CO_2$. The cell density was maintained within a range of $2 \times 10^5$–$1 \times 10^6$ cells/mL. Adherent U87 cells were cultured at 37 °C in DMEM medium supplemented with 10% FBS, 1% P/S, and 5% $CO_2$. Upon reaching confluence, one flask of U87 cells was split into five flasks. Cells were stained with trypan blue and counted using a 2-chip disposable hemocytometer (Bulldog Bio, Portsmouth, NH) to estimate the cell density and viability.

### LPS treatment of HeLa and U87 cells
2 or 4 μL of 2.5 mg/mL LPS were added to the 5 or 10 mL culture media in each HeLa or U87 cell culture flask, to get a final LPS concentration of 1 μg/mL. The HeLa and U87 cells were exposed to LPS for 24 h before being harvested and analyzed.

### Cell pellet collection
HeLa and U87 cells were collected, washed, counted, and subsequently centrifuged into pellets prior to CE-MS analysis. The HeLa cell pellets were obtained by direct centrifugation of the HeLa cell culture suspension at $300 \times g$ for 5 min. To detach the U87 cells from the flask bottom, 0.25% trypsin-EDTA was added, followed by incubation at 37 °C for 5 min. The digestion was stopped by adding complete DMEM medium, and the detached U87 cells were centrifuged at $300 \times g$ for 5 min to obtain the cell pellets. HeLa and U87 cell pellets were washed three times with 1× PBS, and their viability and density were assessed using a 2-chip disposable hemocytometer prior to the final centrifugation at $300 \times g$ for 5 min. The cell viability was typically >90%. The cell pellets were kept on ice until their use.

### Offline cell loading into the CE capillary
One or five cells were loaded offline into the silica surface OptiMS Cartridge, following the protocol described in our previous work[32], with modifications. The cell loading process was visualized and monitored under an IX83 microscope (Olympus, Center Valley, PA), using a 10× magnification. First, the inlet of the CE capillary separation line was immobilized on a glass slide (pretreated with Aquapel®) placed under the microscope. Then, the capillary inlet was immersed in a 40 μL droplet of 1 mM ammonium acetate pH 6.7. A hydrodynamic flow was generated by manually lifting or lowering by ~45 cm the electrospray emitter tip of the capillary (i.e., separation line outlet) to generate an ultra-low flow rate of $48 \pm 7$ pL/s (as determined experimentally, see Supplementary Methods) and enable precise control of the cell influx (the estimated theoretical flow rate was ~74 pL/s, see Supplementary Methods). Flow towards the separation line inlet was created by lifting the emitter tip to expel air bubbles before cell loading or dislodge unwanted cells after cell loading. For cell loading, 5 μL of a cell suspension at ~5 cells/nL was mixed with the droplet in which the separation line inlet was immersed, while the emitter tip of the capillary was held at the same height as the separation line inlet to prevent any forward or backward flow. The cell-containing droplet was gently agitated with a pipet tip until a target single-cell (e.g., with the desired size and morphology) was observed in close proximity to the inlet. Then, the emitter tip of the capillary was lowered to introduce the cell into the capillary, and the flow was maintained until the cell was located approximately 500 μm away from the capillary inlet for targeted cell injection. The same procedure was repeated several times to load manually the desired number of cells.

### Cell staining and visualization
To record the cell morphology and size distribution, 5 μL of a suspension of unstained cells in 1× PBS (with a cell density of ~$1 \times 10^6$ cells/mL) were deposited on a glass slide and imaged with the microscope under bright field at 10× magnification. The average diameters of representative HeLa and U87 single cells were determined to be $21.8 \pm 4.9$ μm (HeLa), and $26.1 \pm 6.6$ μm (U87), respectively, based on selected populations of 564 HeLa cells and 543 U87 cells under the utilized experimental conditions (see Supplementary Fig. 2C, D). Considering the Gaussian cell size distributions, HeLa and U87 cell populations were divided into three subpopulations approximately equal in the number of cells and according to their cell diameters, namely, "smaller," "medium," and "larger" cells. HeLa cells with a diameter ≤19.6 μm and a diameter ≥24.0 μm were considered as smaller and larger HeLa cells, respectively. HeLa cells with a diameter in the range 19.6–24.0 μm were considered as medium size HeLa cells. For U87 cells, the thresholds to determine smaller and larger U87 cells were 23.2 μm and 29.0 μm, respectively. U87 cells with a diameter in the range 23.2–29.0 μm were considered as medium size U87 cells. For improved visualization of the cell morphology and membrane integrity, the cells were stained with CellMask™ plasma membrane green stain, following a procedure adapted from the manufacturer's protocol. The 1000× concentrated stain solution was diluted to 1× working solution with PBS. Subsequently, the cell pellet was resuspended with the working solution to an approximate cell density of $1 \times 10^6$ cells/mL. Then, the cells were incubated in the dark for 30 min, followed by three washes with PBS to remove the excess stain. For fluorescence microscopy imaging of stained cells loaded within the capillary, the polyimide coating was removed before the experiments to avoid interference. To determine the cell viability and membrane integrity under the selected in-capillary sample processing conditions (i.e., after 60 min of incubation with the PNGase F enzyme in 1 mM ammonium acetate pH 6.7 buffer), the cells were stained with LIVE/DEAD™ fixable green dead cell stain. For this, one vial of the fluorescent dye was resuspended with 50 μL of DMSO. HeLa cells were harvested, washed, and resuspended with 1 mM ammonium acetate pH 6.7 to an approximate cell density of $5 \times 10^5$ cells/mL. Then, 1 μL of the resuspended dye was added to 1 mL of the cell suspension. Finally, 20 μL of the stained cells were mixed with 15 mIU of PNGase F. Bright-field and fluorescence-based microscopic images were acquired at different time points to evaluate the cell viability and morphology during the deglycosylation step with PNGase F.

### Preparation and characterization of EV isolate
Plasma-derived EVs were isolated using a size-exclusion chromatography (SEC) column with a Sepharose CL-2B stationary phase. Briefly, 100 μL of platelet-free anticoagulated with EDTA pooled human plasma (from blood donated by self-declared healthy male donors of 23–67 years old) were loaded on the SEC column. EVs were

eluted from the SEC column with 0.1× dPBS, and the EV-containing fractions were pooled. The pooled EV fractions were then concentrated using Amicon® 30 kDa MWCO ultrafiltration devices to a final volume of ~33 μL, and stored at 4 °C until their analysis. The approximate EV particle concentration was estimated to be $1 \times 10^{10}$ EV particles/mL, based on a combination of EV counting, using tunable resistive pulse sensing (TRPS), nano-flow cytometry, and immunoaffinity-based interferometry.

### In-capillary sample processing and CE methods

In-capillary sample processing for N-glycan release with PNGase F and CE-MS experiments were conducted using a CESI 8000™ instrument (SCIEX). In all experiments, bare fused silica (BFS) OptiMS capillaries (91 cm × 30 μm i.d. × 150 μm o.d.) were used. Prior to each online or offline sample injection, a series of rinses of the separation and conductive lines were performed. For the separation capillary, these rinses included: MeOH (100 psi, 10 min), 0.1 M NaOH (100 psi, 3 min), 0.1 M HCl (100 psi, 3 min), and Milli-Q water (100 psi, 5 min), followed by the background electrolyte (BGE) (100 psi, 7 min). The conductive line was rinsed with the BGE (100 psi, 2 min). Before and after online (model glycoproteins, whole plasma, EVs, and ~ten cells (referred to as "bulk cells" in this study)) or offline (1–5 cells) sample loading into the CE capillary inlet, a plug (1 or 2 nL applying 1 psi for 6 or 12 s) of a PNGase F digestion solution at 1.1 mIU/μL in 7 mM NaCl, 3 mM Tris-HCl, and 0.7 mM $Na_2$EDTA was injected into the capillary using the CESI 8000 instrument. For offline cell loading, the CE cartridge was removed from the CESI instrument after the injection of the first PNGase F plug for manual cell loading, as described above, and placed back in the CESI instrument for subsequent in-capillary sample processing. After online or offline sample loading, a short plug of water (1 nL) was injected before the injection of the second PNGase F plug, followed by a short plug (0.5 or 1 nL) of 50 mM ammonium acetate pH 6.7 (see Supplementary Methods). Then, two voltage pulses of 20 kV were applied in normal and reverse polarity for 30 s with the BGE composed of 10 mM (ionic strength) ammonium acetate pH 4.5 with 10% isopropanol, before incubating the capillary inlet in a vial containing 50 mM ammonium acetate pH 6.7 for either 30 min (model glycoproteins, whole plasma, and EVs) or 60 min (mammalian cells). After the in-capillary incubation step (performed at ~12 °C) for N-glycan release with PNGase F, a BGE plug (10 psi for 10 s (model glycoproteins and whole plasma) or 10 psi for 60 s (mammalian cells)) was injected in the capillary prior to label-free CE-MS analysis of released N-glycans performed as described below. All CE methods employed 20 kV in reverse polarity with a voltage ramp time of 1 min. The CE-MS experiments were carried out with a BGE of 10 mM (ionic strength) ammonium acetate pH 4.5 with 10% isopropanol. This BGE generated a relatively low cathodic EOF ($\mu_{EOF}$ $2.02 \times 10^{-8}$ m²/V/s) based on the detection of a neutral marker (acetaminophen). All CE-MS analyses were performed with a CE supplemental pressure (SP) of 5 psi, which was applied 18 min after switching on the CE voltage at the beginning of the CE run. Due to the variability in the manually injected cell plugs (performed offline), the migration time ranges in CE-MS analysis of mammalian cells were normalized, based on the most abundant detected glycan species.

For model glycoproteins (IgM and IgG), isolated from blood serum by size-exclusion chromatography (IgM) and ion-exchange chromatography (IgG), sample injections were performed at 1 or 5 psi for 6 s, corresponding to 1 and 5 nL injection volumes, respectively (i.e., 0.16 and 0.8% of the capillary volume, respectively). Three replicate analyses were performed with the injection of 0.1 ng and 5 ng of IgM, corresponding to ~60 pL and ~3000 pL of human serum, respectively. Three replicate analyses were performed with the injection of 0.5 ng and 5 ng of IgG, corresponding to ~50 pL and ~500 pL of human serum, respectively.

For total plasma isolate, 1 mL of whole blood plasma isolate was centrifuged at 16,000 × g for 20 min at 4 °C, and the supernatant was carefully pipetted to avoid collecting the lipid layer. No protein depletion or enrichment was performed for the total plasma samples (except for a partial removal of lipids, see Supplementary Methods). For CE-MS analysis of total plasma, sample injections were performed at 1 psi for 6 s, corresponding to 1 nL injection volumes, and 5, 50, or 500 pL of plasma, depending on the dilution of the "pre-processed" plasma sample in water.

For EV isolate, sample injections were performed at 1 psi for 6 s (1 nL injected) or 5 psi for 60 s (50 nL injected), corresponding to injected amounts equivalent to ~3 nL and ~150 nL of plasma, respectively.

For cell analysis, the cell pellets were resuspended in 200 μL of 1 mM ammonium acetate pH 6.7 to get a final cell density of ~5 cells/nL. For online cell loading of ~10 cells, 2 nL of a cell suspension at ~5 cells/nL were injected, applying 1 psi for 12 s. As additional verification of the number of injected cells using automated injection, five repetitive analyses were performed with the injection of 2 nL of a cell suspension at ~5 cells/nL, followed by the removal of the cartridge from the CE instrument and cell counting using the microscope-based visualization. The number of injected cells was determined to be $10.5 \pm 4.3$. In the offline cell loading mode, 1–5 cells were selected and injected manually as described above, and the 1–5 cell-containing plugs corresponded to ~4–6 nL injection volumes, based on microscope visualization. Owing to cell size variations, sets of five repetitive analyses were systematically performed with one (offline injection), five (offline injection), and ~ten (online "bulk sample" injection) mammalian cells for each cell type (HeLa and U87 cells). CE-MS analyses of a blank sample of water were performed systematically to confirm insignificant levels of carryover derived from the analysis of preceding biological samples (blood-derived isolates and mammalian cells). For single-cell analysis, a water blank sample was analyzed between each single-cell injection. CE-MS analyses of the cell suspension medium (i.e., 1 mM ammonium acetate cell suspension buffer) were also performed. For these control analyses, 2–4 nL of water or cell suspension medium were injected inside the capillary and processed using the developed workflow, including the digestion step with PNGase F. Careful and thorough rinsing cycles of the capillary were performed with MeOH, NaOH, HCl, and water, as described above, before and after each control analysis. During the method development and optimization stage, CE-MS analyses of single mammalian cells were also evaluated on different days and with different CE capillaries without observing major differences in performance. Unsuccessful measurements, e.g., caused by hardware malfunction (mechanical capillary or emitter damage), were excluded from this proof-of-concept study.

### MS instrumentation and techniques

The CE capillary was interfaced with an Orbitrap™ Fusion Lumos™ mass spectrometer using a Nanospray Flex ion source (both Thermo Fisher Scientific, Bremen, Germany). All analyses were carried out in negative ESI mode. The nanoelectrospray potential was set to −1.8 kV. The ion transfer tube (ITT) temperature was set to 150 °C (the distance between the electrospray emitter tip and the MS ITT was set to ∼5 mm). The CE-MS analyses were performed with automatic gain control (AGC) of $1 \times 10^6$ or 250%, a maximum injection time of 250 ms, 5 microscans, an S-lens voltage set to 65 eV, the nominal resolving power of 120,000 at 200 $m/z$, and in-source collision-induced dissociation (ISCID) at 70 eV. For CE-MS² experiments, instrument resolving power was set at 60,000 at 200 $m/z$ with 1 microscan. AGC was set to $2 \times 10^5$ with a maximum injection time of 1000 ms. An isolation window of 2 $m/z$ was selected, and 32 eV was determined to provide the optimum normalized collision energy. CE-MS¹ was performed as described above.

### Data analysis

For data acquisition and processing, Xcalibur™ (v. 3.1) software was used. CE-MS data were processed with GlycReSoft (v. 3.10) software

(Boston University, Boston, MA, USA). Analyses of CE-MS[2] data were performed with SimGlycan (v. 5.91) software (Premier Biosoft, Palo Alto, CA, USA). The generated results were based on the processing of three replicates (model proteins, total plasma, and EVs) and five repetitive (mammalian cells) analyses. For CE-MS[1] data processing with GlycReSoft, a mass-matching error tolerance of 20 ppm was used in all searches. Up to six charge states and sodium and ammonium adducts were included in the search. Other parameters were the same as described in our previous reports[40,49]. According to the program developers' recommendations[74], only glycan compositions passing the score threshold of 5 were selected (see Supplementary Methods). The glycan identification analysis of the CE-MS data was conducted using database searches against an in-house built mammalian database (version of December 2020) encompassing 27,335 N-glycan compositions (the mammalian database provided with the GlycReSoft software package encompasses 1766 N-glycan compositions). For CE-MS[2] processing with SimGlycan, a 20 ppm precursor mass tolerance, and a 10 ppm fragment mass tolerance were used in all searches. Non-labeled glycans (unmodified or with sodium adduction) were searched selecting the options "Underivatized" and "Free" in the chemical derivatization and reducing terminal windows, respectively. Other parameters were as described in our previous studies[40,49]. The glycan composition identification results were mainly based on CE-MS data processing using GlycReSoft. As additional verification of the plausible glycan composition identifications made using GlycReSoft, several supplementary levels of manual data examination were applied according to our recent studies[40,49]. In brief, this verification included 1. Predictable trends in CE-MS migration patterns with respect to glycan composition, net charge, and molecular mass, 2. Charge state and isotopic distributions characteristic to glycan ions, 3. Detection of neutral losses of monosaccharides (e.g., hexose and N-acetyl-hexosamine), and 4. Manual examination of CE-MS[2] data for low-intensity parent ions. The relative quantitation of the detected N-glycans was based on the single-stage MS signal intensities or peak areas of the detected N-glycans that were normalized with respect to the summed MS signal intensities or peak areas of all the N-glycans detected in the sample. In addition, a qualitative comparison was performed based on the fractional distributions corresponding to the number of specific species (e.g., disialylated glycans) out of the total number of N-glycans detected and identified in the analyzed biological specimens.

The bar charts with individual data points, mean values, and error bars were plotted using the R language and ggplot2 package in the rStudio development environment (2023.03.0 + 386 "Cherry Blossom" Release). The R language in the rStudio development environment was also used to perform statistical one-way ANOVA and two-sided paired $t$-tests. The open-access tBtools-II (v1.120) software[75] was employed to generate heatmap clustering, utilizing the Euclidean distance-based clustering method and the complete cluster approach. For data clustering, N-glycan abundances (based on peak intensities) were normalized with respect to the summed abundances of all the N-glycans detected in the analyzed biological sample. For the blood-derived isolates, the clustering was done using the normalized abundance values after imputing 10% of minimum abundance for missing values followed by $\log_2$ transformation. For the single cells, the clustering was done using $\log_2 (x + 1)$ transformation, where x is the normalized glycan abundance value. The PCA plots were created with the open-access version of ClustVis (https://biit.cs.ut.ee/clustvis/software[76]). The average cell diameters of HeLa and U87 cells were measured using the open-access ImageJ (v1.53k) software. The glycan structures were designed with the open-access version of GlycoWorkBench (v2.0). Other schematic images (e.g., cell structure illustration) were built with the BioRender graphical tool (http://BioRender.com).

For the Euclidean distance-based hierarchical clustering of single HeLa and single U87 cells before LPS treatment, glycans that were detected in at least three CE-MS analyses out of the ten total repetitive

analyses (i.e., five repetitive analyses for single HeLa cells and five repetitive analyses for single U87 cells) were selected. This selection generated a set of 47 glycans highly representative of single HeLa and single U87 cells. For the Euclidean distance-based hierarchical clustering of single HeLa and single U87 cells after LPS treatment, glycans that were detected in at least two CE-MS analyses out of the five repetitive analyses of LPS-treated HeLa cells, and glycans that were detected in at least two CE-MS analyses out of the five repetitive analyses of LPS-treated U87 cells were selected and added to the above-described 47 glycans that are highly representative of HeLa and U87 untreated cells. For the PCA analysis of the glycans identified in single HeLa and single U87 cells after LPS treatment, more stringent parameters were used. In this case, glycans that were detected in at least three CE-MS analyses out of the five repetitive analyses of LPS-treated HeLa cells, and glycans that were detected in at least three CE-MS analyses out of the five repetitive analyses of LPS-treated U87 cells were selected and added to the above-described 47 glycans highly representative of HeLa and U87 untreated cells.

## Reporting summary
Further information on research design is available in the Nature Portfolio Reporting Summary linked to this article.

## Data availability
The raw data generated in this study have been deposited in GlycoPOST (https://glycopost.glycosmos.org) under the accession numbers GPST000378 and GPST000380. Source data are provided with this paper.

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

## Acknowledgements
The authors acknowledge Dr. Marcia Santos from SCIEX for helpful and fruitful discussions. The authors thank Prof. Ionita Ghiran for kindly providing pooled human plasma. Mr. Alan Zimmerman is acknowledged for his excellent help in the preparation of EV samples and fruitful discussions. The authors would like to thank Mr. Somak Ray for his help with building the mammalian glycan database and Dr. Getulio Oliveira for helpful recommendations related to cell culture and cell staining experiments. This work was supported by the National Institutes of Health under the award numbers R01CA218500 (ARI) and R35GM136421 (ARI). The authors acknowledge Thermo Fisher Scientific for their support through a Technology Alliance Partnership program (ARI) and SCIEX for their support and fruitful discussions.

## Author contributions
A.L.M. and Y.G. developed the integrated platform for the profiling of cell surface N-glycans, performed the experiments, and processed and interpreted the data. A.L.M. developed and optimized the CE-MS methods and performed the analysis of blood-derived isolates. Y.G. developed and optimized the in-capillary cell loading process, performed cell culture experiments, and acquired microscopy images. A.R.I. conceived the experimental concept and guided the implementation of the experimental plan and the overall study. A.L.M., Y.G., and A.R.I. designed the study, evaluated and interpreted the results, and wrote the manuscript. All co-authors read and commented on the manuscript and supplementary information.

## Competing interests
The authors declare no competing interests.
