## [Peer Review File · Nature Communications]

Reviewers' Comments:

Reviewer #1:

Remarks to the Author:

In this manuscript, the authors have developed a combined workflow of online in-capillary sample processing with label-free CE-MS for single-cell N-glycomics from mammalian cells. Nearly 100 N-glycans was detected per single cell from HeLa and U87 cells as proof-of-concept and this workflow was applied to plasma samples that are more complex. The manuscript is detailed but there is a lot of repetitive information which can be eliminated and make the manuscript more concise – please also address the comments below:

1. Was any depletion or enrichment done for the plasma samples?
2. How is the loss of sample accounted for when dealing with such low amounts?
3. It is unclear if the workflow was optimized based on samples such as IgM and IgG or the amounts/mol were consistent irrespective of sample type?
4. Clarify if the ~7 fold increase in N-glycan identification is based on single cell analysis between this study and other studies?
5. Please address relative abundances of major glycan classes in addition to the sialylation/fucosylation levels for all sample types.
6. Also put a range for "high" and "medium" abundances of glycans.
7. It will be easier to follow the numbers/amounts of injection vs N-glycan identification in a table format or in supplementary. It is often getting confusing in the text with so many different numbers.
8. Does all identified glycans including the heavily branched structures have MS/MS evidence? Are all the cross-ring fragments identified determined by this method or taken from existing studies?
9. Is the CE-MS approach biased towards any class or particular glycan structure (other than the heavily branched ones) – please clarify
10. For EVs, why is the comparison in such a larger range between 3 and 150 nl?
11. Need to increase size of labels for all figures – very difficult to read
12. Figure 2A: do the bar graph colours mean anything?
13. What is the reason for comparing between single, 5 and 10 HeLa cells when this is not applied to any other sample types?
14. Based on figure 4K, why is comparing HeLa and U87 glycome and overlap relevant?
15. There is a lot of information present, but it is often convoluted, and the main take home message is getting lost in certain sections such as for EVs and plasma was it really a single cell analysis or was it minute amounts/volumes, what is total abundance of oligomannose, complex, hybrid etc for each of the sample types, was any proteomics done and any improved identifications were found?

Reviewer #2:

Remarks to the Author:

The manuscript presents a proof-of-concept study of a very interesting and potentially highly beneficial innovation. It combines several principal advantages of capillary electrophoresis, which enables work with single cells or extremely small sample volumes, allows for mixing different zones in the capillary, and enables performing reactions, such as N-glycan release using PNGase F, directly in the capillary. Another virtue of capillary electrophoresis and the presented methodology is that whole intact cells can be introduced in the capillary and only surface N-glycans can be cleaved off.

The authors have shown that their novel approach has great potential in N-glycan profiling of single cells, small numbers of cells, or extremely small volumes of blood samples. They were able to identify a significant number of glycan structures. What is more, they managed to clearly distinguish the changes in N-glycan profiles of lipopolysaccharide-stimulated cells, proving thus the applicability of their methodology to comparative studies, aiming to evaluate the influence of various factors on N-glycan profiles.

In my view, the manuscript presents noteworthy results that are of great significance to the fields of glycomics, glycoproteomics, disease biomarker discovery, etc. The conclusions are supported by

experimental results that are appropriate for a proof-of-concept type of study. The experiments are described in detail and clearly.

Considering the above I can recommend the manuscript for publication. I just suggest the authors add two minor details to the description of the experiments:

1) Several times, injected volumes or flow rates during the hydrodynamic sample introduction are mentioned. Were these volumes/flow rates calculated based on Poiseuille's law or was the flow rate measured in some way? Both approaches, calculation and measurement, have their limitations and can suffer from certain inaccuracies. It would be helpful to know which approach the authors chose and what were their input data for calculation, or the experimental setup for flow rate measurement.

2) Page 6, line 247 - I would like to know why a short plug of water was injected after introducing the sample, before injecting the PNGase F solution. I would appreciate a brief comment on this in the text.

Overall, I can only congratulate the authors of this exciting work.
Tomas Krizek

Reviewer #3:

Remarks to the Author:

In their study Marie et al. present the first unbiased and label-free glycomics study on single cell level using CE-MS. They identify around 15 and 60 N-glycans from the cell surface of HeLa and U87 cells respectively while maintaining the integrity of the cells. This is of high potential interest as it could allow the combined analysis with the proteome of the very same cell in future. The authors further apply their sensitive method to ultra-low input amounts of extracellular vesicles and plasma isolates.

The technical challenges taken here can also be of help for other ultra-low input applications, not only limited to glycomics studies. They will likely be further improved with newer generation MS instrumentation. That said, this reviewer is convinced that this pioneering SCG study will be of high interest to the community, however some concerns and questions should be addressed as detailed below.

Comments

- In this study IgM and IgG were used as model glycoproteins. At the lower end 0.1 ng IgM and 0.5 ng IgG were used. I was wondering why different amounts of protein were selected for the comparison of IgM and IgG. The method section states to this that those inputs correspond to 60 and 50 pL of commercial human serum respectively (again not the same amount, but this might of course vary dependent on the source).

- As described, cells were squeezed in the capillary when the cell-size was close to its inner diameter and this effect was used to facilitate cell stacking.
 - o What was the exact inner diameter of the capillary, and did it correspond to the average size of a HeLa/U87?
 - o Did you observe problems with stacking for smaller cells, or were they still sticky?
 - o How can the chosen workflow be adapted to other cell-types, especially larger ones (I guess one would need dedicated capillaries for each cell type, but very heterogenous populations might be problematic?)

- Figure 3A-C): The profile of 1x HeLa looks very different to 5x or 10x cells and looks way more symmetrical. Do you have an idea why and have you compared profiles from single cells between replicates as well?

- In Figure S3A the variance of raw abundance of glycans from single HeLa seems even lower than that of bulk replicates – how does this agree with the described (and expected) higher variance of glycan abundance in single cells (lines 615-618)?

- As already highlighted by the authors themselves, it is surprising that the U87 cells yielded much more glycosylation than expected when comparing them to HeLa even when considering their size difference. It might be important for future biological studies to ensure that all or at least close to all glycans are successfully released upon digestion to get a proper fingerprint of each cell. As already suggested in line 683-685 of the manuscript this might not be the case for HeLa. To check one could repeat the analysis with an alternative glycosidase or chemically release glycans instead (e.g. using NaOCl, doi: 10.7171/jbt.19-3004-001)

- This work is a very exciting first step into performing unbiased glycomics at the single cell level and will be important for future biological studies as well. With this in mind, I was wondering if you can comment on scalability? Would it be possible to do this in a high throughput manner i.e. 100 or more cells in a reasonable time (can it be automatized, how many cells can be stacked into one capillary,...?)

- I tried to check the data on GlycoPOST but got a "No object registered" prompt when searching for the dataset GPST000378. Is it publicly available or do I need a password (ahead of publication)?

Minor Comments

- Line 418 seems to be grammatically wrong/ a typo: Suggestion: "We noticed that the development and optimization of the workflow as well as its adoption to..."
- Fig 3H-K : The annotations are very hard to read, upon zooming, the quality is not good enough to properly read the numbers in the Venn diagrams. Maybe those panels can be moved to a separate figure or at least somehow enlarged.
- Line 624: I guess this should be referenced to Figure 4I (not 4E)
- General: When reading the manuscript, it seems there is a lot of jumping up and down between figures which lowers smooth readability. I'd recommend restructuring a bit to ease readability. (E.g., figure 4J and K is mentioned in the text the first time after figure 6 description and could be moved down.)

Point-by-point responses to Reviewers' comments

Title: In-capillary sample processing coupled to label-free capillary electrophoresis-mass spectrometry to decipher the native N-glycome of single mammalian cells and ng-level blood isolates

Authors: Anne-Lise Marie, Yunfan Gao, and Alexander R. Ivanov

The authors thank the Reviewers for carefully evaluating our manuscript and for providing positive feedback and helpful and constructive comments. We are happy that our proof-of-concept study for label-free CE-MS analysis of limited amounts of biological samples and single mammalian cells was found - very interesting and of potentially highly beneficial innovation; our experiments and results - clearly presented, and our efforts in developing a sensitive CE-MS technique for direct and unbiased N-glycan profiling of small populations of cells and single cells were well recognized. We addressed the concerns and comments made by the Reviewers in full. We strongly believe that the revised version of the manuscript was significantly improved with the substantial and careful changes that we made based on the Reviewers' suggestions.

Please see below our point-by-point responses in blue font.

Reviewer #1:

In this manuscript, the authors have developed a combined workflow of online in-capillary sample processing with label-free CE-MS for single-cell N-glycomics from mammalian cells. Nearly 100 N-glycans was detected per single cell from HeLa and U87 cells as proof-of-concept and this workflow was applied to plasma samples that are more complex. The manuscript is detailed but there is a lot of repetitive information which can be eliminated and make the manuscript more concise – please also address the comments below:

We acknowledge Reviewer 1 for their thorough examination of our manuscript and for providing valuable and thoughtful comments. We provided additional information and eliminated repetitions in the revised version to address the comments made by this Reviewer.

1. Was any depletion or enrichment done for the plasma samples?

As mentioned in the revised manuscript (page 18), in this study, we analyzed platelet-free anticoagulated with ethylenediaminetetraacetic acid (EDTA) pooled total human plasma from blood samples collected from self-declared healthy donors. The total plasma isolates were prepared using differential centrifugation, as adapted from previously published work (PMID: 26745887). Briefly, human blood was collected in EDTA-containing vacutainer tubes. Then, total blood was centrifuged at 500 x g for 10 min to obtain total plasma, followed by two plasma centrifugations:

1,000 x g for 10 min and 10,000 x g for 10 min at room temperature to remove cell debris. Plasma was centrifuged at room temperature to prevent activation of platelets. Prior to CE-MS analysis, the total plasma isolate was centrifuged at 16,000 x g for 20 min at 4 °C, and the supernatant was carefully pipetted to avoid collecting the lipid layer. As indicated on page 21 of the revised manuscript, the total plasma isolate was diluted in water for sample injection in the CE capillary. Therefore, no protein depletion or enrichment was performed for the plasma samples (except for a partial removal of lipids, as indicated above) unless specified otherwise for other described blood-derived isolates. For the sake of clarity, additional information was provided in the revised manuscript (pages 18 and 21, and Supplementary Methods).

2. How is the loss of sample accounted for when dealing with such low amounts?

We thank the Reviewer for the helpful comment. The whole point of injecting and processing intact cells or minimally processed, or unprocessed samples in the capillary was to eliminate or at least substantially minimize sample losses caused by various reasons. We think that in our proof-of-concept (POC) study, the vastly simplified in-capillary sample preparation approach coupled online with CE-MS mostly eliminates sample losses associated with sample handling and transfer steps in the offline approach. N-glycans are directly released from the surface of mammalian cells (injected in a controlled manner by hydrodynamic pressure and microscopy-based visualization) or from blood-derived isolates inside the capillary for their subsequent online CE-MS analysis. Consequently, there should be virtually no sample loss (or minimal loss) at this stage of the developed workflow. Nevertheless, trace amounts of released glycans may bind to the inner capillary wall and may still lead to some decrease in the CE-MS signal. It is difficult (if not impossible) to assess sample losses derived from the adsorption of trace amounts of glycans on the silica surface. Nevertheless, CE-MS analyses performed with the injection of a water blank sample allowed us to show that the levels of carryover derived from the analysis of preceding biological samples were insignificant (see Figure 3EG, pages 5, 9, and 26 of the revised manuscript). These results demonstrated the reliability of the number of glycans identified in the biological samples analyzed in this study and that run-to-run sample losses were negligible or below the limit of detection (LOD) of the developed CE-MS method. In the field of single-cell analysis, a significant challenge is the detection sensitivity of MS instruments (or other types of detectors). With the newer generation of mass spectrometers with further improved sensitivity, we may be able to better assess the sample loss associated with our in-capillary sample processing workflow and, at the same time, further increase the number of detected glycans using extremely low amounts of biological samples, including single cells, organelles, and microvesicles. We look forward to further developments in the field of mass spectrometry! Based on the Reviewer's comment, additional details were provided on pages 9 and 26.

3. It is unclear if the workflow was optimized based on samples such as IgM and IgG or the amounts/mol were consistent irrespective of sample type?

We thank the Reviewer for making this point. In our study, IgM, a heavily glycosylated protein, was selected to develop and optimize the in-capillary sample processing method for N-glycan release coupled online to label-free CE-MS analysis. We thereafter applied the optimized CE-MS-based workflow to other types of blood-derived isolates (IgG, total plasma, and total extracellular vesicle (EV) isolates) and, finally, to more challenging biomedically-relevant samples, i.e., small

populations of cells and single cells. The injected amounts of biological samples were modified and optimized depending on the concentration of the specific analyte type and the glycosylation level in the corresponding biological sample (see also our response to point 10). For IgM and IgG, the lowest injected amounts (0.1 ng and 0.5 ng, respectively), corresponding to the total of ~10 pg of N-glycans for each protein, were equivalent to the amounts of IgM and IgG isolated from ~60 pL and ~50 pL of human serum, respectively (according to our estimations based on the literature data). These lowest injected amounts were selected for a comparative analysis with the injection of ~50 pL of total human plasma (see Figure 2A), and in order to release amounts of glycans corresponding, in theory, to the glycan content of one single mammalian cell (i.e., ~10-50 pg of glycans/single cell, on average, PMID: 34253696). For the total EV isolate, the highest injected amounts (i.e., 50 nL of the EV isolate equivalent to the EV content of ~150 nL of plasma) and the lowest injected amounts (i.e., 1 nL of EV isolate equivalent to the EV content of ~3 nL of plasma) were selected based on the acquired MS signal intensity levels. Using these selected high and low amounts of the EV isolate, the MS signal intensity levels of detected glycans in CE-MS analysis of the EV isolate were comparable to the signal levels recorded in CE-MS analysis of ~50 pL and ~5 pL of total plasma, respectively (see also point 10 below). Additional details were provided in the supporting information of the revised manuscript as Supplementary Note 1 (see pages 4 and 6).

4. Clarify if the ~7 fold increase in N-glycan identification is based on single cell analysis between this study and other studies?

In this study, the developed in-capillary sample processing workflow coupled online to CE-MS resulted in the ~7-fold increased numbers of N-glycans identified in IgM, IgG, total plasma, and total EV isolates, using sub-0.5 ng-levels of serum proteins and nL/pL-levels of plasma isolates, in comparison to other reported N-glycan profiling studies focused on N-glycan profiling of blood-derived isolates of IgM, IgG, plasma, or EVs. As stressed out below (see point 13), the direct comparison of the numbers of glycans detected in single mammalian cells using our developed CE-MS workflow with published data remains challenging, if not impossible, owing to the current lack of analytical technologies enabling direct and unbiased glycan profiling of single cells. Additional clarifications were provided in the revised manuscript (page 15).

5. Please address relative abundances of major glycan classes in addition to the sialylation/fucosylation levels for all sample types.

As emphasized in our answer to point 8 and also stressed out by this Reviewer (see below), the accurate and unambiguous characterization of the structure of all identified glycans requires tandem MS analysis. Consequently, providing abundances of complex-, hybrid-, and high-mannose-type N-glycans detected in the biological samples analyzed in this study would require the full structural characterization of all the identified by CE-MS¹ glycans, based on CE-MS² experiments. In our presented work, the conducted POC CE-MS² experiments, using the MS equipment we currently have access to in the lab, expectedly did not result in the structural characterization of all N-glycans detected and identified in CE-MS¹ analyses. This may be explained by 1- the stochastic mode of MS² data dependent acquisition; 2- the extremely low abundances of some precursor glycan ions (in this case, the acquired MS² spectra were insufficiently informative for reliable and accurate structural characterization); and 3- the

limitations of the currently available glycan-dedicated software, glycan databases, and spectral libraries (see point 8 below). In the revised manuscript, we provided the relative abundances (i.e., abundancies based on the normalized N-glycan signal intensities) of all N-glycans detected and identified in the CE-MS¹ analyses of the blood-derived isolates and mammalian cells (see Supplementary Tables 2 and 5). We also provided the relative distributions of complex-, hybrid-, and high-mannose-type N-glycans characterized in the analyzed mammalian cells using our POC CE-MS² method (page 10). As highlighted in the introduction of our manuscript (page 2) and in our previously reported manuscript (PMID: 36959283), alterations of sialylation and fucosylation levels were reported in human pathologies and evaluated as potential disease biomarkers. Also, novel therapeutic drugs or vaccines, which are currently in clinical evaluation, target tumor-associated carbohydrate antigens, like polysialic acids (PMIDs: 30858582, 33462432, and 35158915). We, therefore, think that it is crucial to determine the fractional distributions and abundances of sialylated and fucosylated glycans. This information can be acquired in a straightforward way using our developed label-free CE-MS¹ method. Our future method development efforts will be focused on the structural characterization of larger numbers of N-glycans released from small populations of cells and single cells, which will allow us to accurately determine the abundances of the three major classes of N-glycans.

6. Also put a range for “high” and “medium” abundances of glycans.

As suggested by the Reviewer, we included in the revised manuscript the ranges of abundances corresponding to “high” and “medium” abundances of detected N-glycans for all types of samples analyzed in this study (blood-derived isolates and mammalian cells). For the four types of analyzed blood-derived isolates, the ranges of log₂ values of the normalized signal intensities corresponding to high, medium, and low abundances were 6 to -3, -3 to -6, and -6 to -12, respectively. The log₂ values lower than -12 corresponded to glycans that were not detected in the analyzed blood-derived isolates (see Methods section, page 22, Figure 2J, and Supplementary Table 2). For mammalian cells, the ranges of log₂ values corresponding to high, medium, and low abundances were 6 to 3, 3 to 2, and 2 to 0.04, respectively. The log₂ values equal to 0 corresponded to glycans that were not detected in the analyzed mammalian cells (see Methods section, page 22, Figure 6AC, and Supplementary Table 5AB).

7. It will be easier to follow the numbers/amounts of injection vs N-glycan identification in a table format or in supplementary. It is often getting confusing in the text with so many different numbers.

In the revised manuscript, several figures (Figures 2A, 3G, and 4J) alongside the narrative assist the readership in following the numbers of N-glycans that were detected in each sample type, with the information on the corresponding injected sample amounts. The mean values of identified N-glycans, the individual data points corresponding to the replicate analyses, and the error bars corresponding to the standard deviations (SD) are also provided in these figures. We think that these figures, where the bar plots corresponding to a specific blood isolate are colored with the same base color (e.g., the bar plots corresponding to the IgM isolate are colored with two different purple tones, depending on the injected concentration), appropriately highlight the numbers of identified N-glycans, while making the comparison easier between the numbers of N-glycans identified in the analyzed biological samples (blood-derived isolates and mammalian cells) (see

also our response to point 12). In addition to the described modifications, we provided Supplementary Table 1 in the revised manuscript (see Supplementary Materials).

8. Does all identified glycans including the heavily branched structures have MS/MS evidence? Are all the cross-ring fragments identified determined by this method or taken from existing studies?

We thank the Reviewer for their valuable comment. Tandem MS is indeed required to accurately and unambiguously characterize the structure of all identified glycans, including heavily branched glycans. Our POC CE-MS² experiments resulted in the structural characterization of >60 N-glycans detected in the mammalian cells analyzed in this study, among which highly branched and highly sialylated glycans (see Supplementary Tables 3 and 6). Characteristic CE-MS² spectra of complex-type and high-mannose-type glycans are presented in Figure 5 and Supplementary Figure 4, which display the most intense/relevant fragments derived from the cross-ring fragmentation of the afore-mentioned glycans, using the label-free CE-MS² method developed in this study. All the glycan structures depicted in Supplementary Tables 3 and 6 are based on the CE-MS² spectra acquired in the CE-MS² analyses of the mammalian cells selected in this study. The glycosidic, glycosidic/glycosidic, cross-ring, and cross-ring/glycosidic fragments detected in the acquired CE-MS² spectra were identified based on data processing with SimGlycan[®] software or manual examination. Our developed CE-MS² method allowed us to structurally characterize ~60% of the N-glycans detected and identified in the CE-MS¹ analyses of HeLa cells. Approximately 40% of N-glycans identified in the CE-MS¹ analyses of HeLa cells could not be structurally characterized by CE-MS², most probably due to their very low abundances, or under-representation in HeLa cells (see also our answer to point 5). We think that with the development of 1- newer generation of mass spectrometers with improved sensitivity and duty cycle, 2- smart on-the-fly MS data acquisition technologies to target glycans for subsequent fragmentation using alternative techniques (e.g., UVPD, ECD, ETD, etc.), 3- powerful and automated glycan-dedicated software capable of accurately and unambiguously assigning fragment ions derived from multiple internal fragmentation cleavages, and 4- expanded glycan databases and spectral libraries, encompassing larger varieties of highly/heavily branched and/or highly/heavily sialylated/fucosylated glycan structures, we certainly will be able to dramatically increase the numbers of glycan structures fully characterized in our CE-MS² analyses of mammalian cells or other highly complex and heterogeneous amount-limited biological samples. At this stage of the single-cell glycomics (SCG) field development, the authors are satisfied with the structural characterization of ~60% of glycans that are identified in HeLa cells, based on the injection of only a few HeLa cells.

To address the Reviewer's comment, we expanded the discussion related to CE-MS²-based glycan structural characterization results described in this study and provided Supplementary Note 3 in the revised manuscript (see pages 10 and 16).

9. Is the CE-MS approach biased towards any class or particular glycan structure (other than the heavily branched ones) – please clarify.

We appreciate the thoughtful question. In our presented CE-MS approach, N-glycans are analyzed in their native underivatized state to preserve their integrity and endogenous glycan features (i.e., preserve their native monosaccharidic composition and structural features, including antenna-

branching and glycosidic linkages) and to eliminate the drawbacks and biases associated with any labeling procedures, including incomplete derivatization, side-products, sample losses during cleanup steps, MS signal interferences by the labeling reagent, and high levels of defucosylation/desialylation during sample preparation and MS analysis (see also our recently published manuscript, PMID: 36959283). Therefore, our developed label-free CE-MS method mostly eliminates any sample processing-related analyte alterations and additional sources of pre-analytical variability in glycan identification and quantitation using minute amounts of biological samples because the in-capillary sample preparation approach is vastly simplified, and the glycans are analyzed in their native underivatized state. In addition, conversely to lectin-based analytical technologies, which result in undirect and biased profiling of certain cell-surface glycans (i.e., the glycans are not released from the cell surface for their direct detection and quantification, and the glycan detectability is highly dependent on the lectin binding affinity and the efficiency of binding, which might be influenced by various factors, including the binding site accessibility and steric hindrance), our developed CE-MS method allows direct and unbiased characterization and quantification of the N-glycomes of mammalian cells or other amount-limited biological samples. For all these reasons, our developed workflow mostly eliminates any potential sources of analytical bias in the detection and quantification of glycans. The developed label-free CE-MS method allowed us to separate, detect, identify, and structurally characterize negatively-charged (i.e., sialylated glycans) as well as neutral glycans (e.g., high-mannose-type glycans), with a large variety of glycan structural features, from simple bi-antennary structures to highly/heavily branched structures. Besides, we showed in this study that our developed method resulted in ~7-fold increased numbers of N-glycans identified in the selected blood-derived isolates, in comparison to the literature data (see also our response to point 4), and in the detection of glycan species with high levels of sialylation and/or fucosylation. Based on these results, we believe that our label-free CE-MS method is capable of accurately recapitulating the complexity of glycans present in the analyzed biological samples and highly minimizes potential analytical bias. Based on the Reviewer's concern, clarifying details were included in our revised manuscript (pages 10, 15-17).

10. For EVs, why is the comparison in such a larger range between 3 and 150 nL?

The injected amounts of biological samples were adapted depending on several criteria (protein concentration, glycosylation level of the analyzed model glycoproteins, concentration of EV particles in the plasma-derived isolate) in order to make a differential comparative analysis of the four types of blood-derived isolates (IgM, IgG, total plasma and EVs), using similar equivalent injected amounts of plasma or serum (see our response to point 2). For the total EV isolate, the highest injected amounts (equivalent to the EV content of ~150 nL of plasma) and the lowest injected amounts (equivalent to the EV content of ~3 nL of plasma) were based on the acquired MS signal intensity levels. Using these selected high and low amounts of the EV isolate, the MS signal intensity levels of detected glycans in the EV isolate corresponded to the signal levels recorded in the CE-MS analysis of ~50 pL and ~5 pL of total plasma, respectively. The injected EV sample amounts were also selected for a comparative analysis with the results acquired with our previously reported label-free CE-MS method (PMID: 36959283) (see page 6 of the revised manuscript). Additional details were provided in the supporting information of the revised manuscript as Supplementary Note 1 (see pages 4 and 6).

11. Need to increase size of labels for all figures – very difficult to read.

Addressed in the revised manuscript.

12. Figure 2A: do the bar graph colours mean anything?

In Figure 2A, we selected a different color for each analyzed blood-derived isolate to help the readership, alongside the narrative, follow the numbers of N-glycans that are detected in each sample type, with information on the corresponding injected sample amounts. The colors have been selected randomly, but we used the same base color for one dedicated blood isolate (e.g., the bar plots corresponding to the IgM isolate are colored with two different purple tones, depending on the injected concentration). We used the same “color codes” throughout all the panels of Figure 2. See also our response to point 7.

13. What is the reason for comparing between single, 5 and 10 HeLa cells when this is not applied to any other sample types?

In simple terms, the reason for including 5- and 10-cell measurements in addition to single-cell glycan profiling was to prove experimentally that the data acquired at the single-cell level do make sense. As emphasized in the revised manuscript (pages 2-4, and 14-15), SCG is an emerging and highly challenging field. So far, methods enabling the direct analysis, characterization, and quantitation of cell glycomes at the single-cell level have not been reported yet. Using lectin-based analytical technologies, glycan detectability is highly dependent on lectin binding affinity, specificity, and steric accessibility, and some cell surface glycans may not interact with the lectins selected in the previously reported methodologies. Therefore, the comparison of our results with published data remained challenging, if not impossible (see also point 4). In order to provide reliable glycan identification results in the CE-MS analysis of single HeLa cells and confirm the increased depth of N-glycome profiling at the levels of 5 and 10 HeLa cells, CE-MS experiments were performed with the injection of single HeLa cells as well as small populations of HeLa cells (i.e., 5-10 cells). These CE-MS experiments aimed to conduct a qualitative and quantitative comparison between the N-glycan profiling of one single cell, five, and ~ten cells, and, as expected, they confirmed a good agreement between the numbers of injected cells, the numbers of detected glycans, and the corresponding peak intensities/areas (as well as the expected cell-to-cell and single-cell to larger cell count variability in glycan profiles). Supplementary Figure 3 shows the linear relationship between the injected cell numbers and the quantified total cellular glycan amounts for eight selected representative glycans. CE-MS analyses were also performed with the cell suspension medium to confirm that the glycans detected in one single HeLa cell were not derived from extracellular proteins or contaminants present in the cell medium. The glycan profiling of larger numbers of HeLa cells (i.e., 5-10 cells) provided additional information about the cell surface glycome, which allowed us to pinpoint the types of glycans that may be present at extremely low abundances at the surface of HeLa cells. Thus, specific glycans (e.g., highly sialylated and/or fucosylated) were detected at low abundances in 5-10 HeLa cells but were not detected at the single-cell level (see Figure 4AB and pages 9 and 16). Further development of our POC study will be focused on improving the sensitivity of detection in an attempt to detect and characterize extremely low abundance glycans at the single-cell level (such glycans could serve as potential disease biomarkers and drug targets; see our answer to point 5). Based on the results

presented in the manuscript, we expect to observe the increased depth of N-glycome profiling at the levels of five and ten (or even more) U87 cells and potentially other cell lines (see also our response to point 6 of Reviewer 3 below). However, examining these trends experimentally was not the focus of this POC study.

To address the Reviewer's comment, additional details were provided on pages 11 and 15 of the revised manuscript.

14. Based on figure 4K, why is comparing HeLA and U87 glycome and overlap relevant?

We thank the Reviewer for their valid question. The main purpose of the experiment was to conduct a preliminary assessment of whether the developed technique can help differentiate cell types and states based on the characterized glycome. Throughout the narrative, several figures (Figures 3H-L and 4K, and Supplementary Figure 5AB) show the overlaps of the glycans identified in HeLa and U87 cells, with and without LPS treatment. This qualitative glycan composition differential analysis allowed us to pinpoint which types of glycans were uniquely detected in the analyzed biological samples, and, on the other hand, which glycans were commonly detected in several sample types. In the field of biomarker discovery, glycomedicine, or more fundamental biology applications, glycan profiling techniques target, inter alia, unique and specific glycans overexpressed in mammalian cells or other types of biological specimens. The characterization of glycosylation profile alterations between different cell lines and phenotypes, and induced by the treatment of cells with biological and chemical agents, relies on different criteria, including the numbers and abundances of identified glycans as well as their monosaccharide compositions. As highlighted above (see point 5), information on the monosaccharide compositions of the detected glycans, which enables the assessment of the sialylation and fucosylation levels, can be acquired in a straightforward way using our developed technique. Besides, as stressed out in pages 13-14, LPS treatment of U87 cells did not induce significant changes in the fractional distributions and abundances of sialylated glycans but, interestingly, induced a modification in the type of biosynthesized sialylated glycans in U87 cells. Such glycosylation subtlety can only be revealed with a thorough examination and comparison of the glycan compositions identified in the different sample types. We, therefore, deemed it relevant to systematically show the overlaps between the glycan compositions identified in the biological samples analyzed in this study.

According to the Reviewer's question, additional details were provided in pages 13 and 15 of the revised manuscript.

15. There is a lot of information present, but it is often convoluted, and the main take home message is getting lost in certain sections such as for EVs and plasma was it really a single cell analysis or was it minute amounts/volumes, what is total abundance of oligomannose, complex, hybrid etc for each of the sample types, was any proteomics done and any improved identifications were found?

We think that most of the concerns raised by the Reviewer here were addressed in points 3, 5, and 7 above. Our study was articulated in two distinct parts. One is related to the analysis of blood-derived isolates (IgM, IgG, total plasma, and EV isolates), and the second one is to small populations of cells and single cells. In the revised manuscript, we improved the organization of the paper, where subheadings were introduced alongside the narrative to clearly differentiate the

sample types analyzed in this study and the corresponding results. The work we presented here was focused on glycomic analysis of limited amounts of biological samples and single cells. Therefore, we did not perform proteomics analysis of the biological samples selected in this study. Nevertheless, as highlighted in pages 15-16 of the revised manuscript, the POC SCG approach allowed us to preserve cell integrity during the enzymatic release of glycans, which may potentially benefit the multi-omic (including proteomic and spatial proteomic) characterization of individual cells. The discussion section of the revised manuscript was expanded according to the Reviewer's comment (pages 15-16). Finally, we would like to mention our recently published article (PMID: 36194750) related to CE-MS-based top-down proteomics of single cells (including HeLa cells).

Reviewer #2:

The manuscript presents a proof-of-concept study of a very interesting and potentially highly beneficial innovation. It combines several principal advantages of capillary electrophoresis, which enables work with single cells or extremely small sample volumes, allows for mixing different zones in the capillary, and enables performing reactions, such as N-glycan release using PNGase F, directly in the capillary. Another virtue of capillary electrophoresis and the presented methodology is that whole intact cells can be introduced in the capillary and only surface N-glycans can be cleaved off.

The authors have shown that their novel approach has great potential in N-glycan profiling of single cells, small numbers of cells, or extremely small volumes of blood samples. They were able to identify a significant number of glycan structures. What is more, they managed to clearly distinguish the changes in N-glycan profiles of lipopolysaccharide-stimulated cells, proving thus the applicability of their methodology to comparative studies, aiming to evaluate the influence of various factors on N-glycan profiles.

In my view, the manuscript presents noteworthy results that are of great significance to the fields of glycomics, glycoproteomics, disease biomarker discovery, etc. The conclusions are supported by experimental results that are appropriate for a proof-of-concept type of study. The experiments are described in detail and clearly.

Considering the above I can recommend the manuscript for publication. I just suggest the authors add two minor details to the description of the experiments:

The authors acknowledge Dr. Krizek for his positive and enthusiastic feedback, and for providing thoughtful and helpful comments. We hope that this Reviewer will find the additional experimental details provided in the revised manuscript helpful.

1) Several times, injected volumes or flow rates during the hydrodynamic sample introduction are mentioned. Were these volumes/flow rates calculated based on Poiseuille's law or was the flow rate measured in some way? Both approaches, calculation and measurement, have their limitations and can suffer from certain inaccuracies. It would be helpful to know which approach the authors chose and what were their input data for calculation, or the experimental setup for flow rate measurement.

We thank the Reviewer for bringing up this question. The volumes of the liquid solutions (blood isolates, cell culture medium, water, BGE, etc.) injected in the CE capillary by hydrodynamic

pressure, using a CESI 8000™ instrument, were estimated based on the Hagen-Poiseuille equation (eq. 1). For the offline loading of one or five mammalian cells into the CESI OptiMS™ cartridge, a hydrodynamic flow was generated by manually lowering by ~45 cm the electrospray emitter tip of the capillary (i.e., the separation line outlet). The estimation of the theoretical hydrodynamic flow rate was based on the Hagen-Poiseuille's law (eq. 1) and eq. 2.

$$Q = \frac{\Delta P \cdot \pi \cdot d^4 \cdot t}{128 \cdot \eta \cdot L} \quad (\text{eq. 1})$$

Where Q represents the volumetric flow rate through the CE capillary; ΔP is the pressure difference between the inlet and outlet ends of the CE capillary (ΔP is equal to 4.3×10^3 Pa, based on eq. 2); d is the diameter of the capillary (i.e., 3×10^{-5} m); L is the length of the capillary (i.e., 9.0×10^{-1} m); and η is the viscosity of the BGE ($\sim 1.3 \times 10^{-3}$ Pa·s).

$$\Delta P = \rho g h \quad (\text{eq. 2})$$

Where ρ is the density of the BGE (i.e., 9.86×10^2 kg/m³); g is the gravity of earth (i.e., 9.8 m/s²); and h is the height difference of the BGE (i.e., 4.5×10^{-1} m).

Based on equations 1 and 2, the theoretical hydrodynamic flow rate was $\sim 7.4 \times 10^{-14}$ m³/s (i.e., ~ 74 pL/s).

We also determined experimentally the hydrodynamic flow rate generated by manually lowering the emitter tip by ~45 cm. For this, we used microscopy-based visualization to determine the laminar flow velocity, based on the assessment of the time required for an air plug to flow from point A to point B through a capillary filled with the BGE (i.e., 10 mM ammonium acetate pH 4.5 with 10% isopropanol). Briefly, one end of a bare fused silica capillary (30 μm ID x 150 μm OD x 90 cm length) was immobilized on a glass slide, placed under a microscope. To fill the capillary with the BGE, the other end of the capillary was inserted into a pressurized vial filled with the BGE. After the entire capillary was filled with the BGE, the capillary was removed from the pressurized vial, and the free end of the capillary was lowered by ~45 cm to generate a height difference between both ends of the capillary. Driven by the height difference, the liquid in the capillary flowed towards the free end and generated an air plug at the immobilized end. The air plug was then trapped by adding a droplet of BGE to the immobilized end. Using the air plug as an indicator, the time required for the air plug to flow through a length of ~1.1 mm was recorded. The average flow velocity was calculated based on the exact length and flow time recorded in three replicate experiments. The determined laminar flow velocity was $6.7 \pm 1.0 \times 10^{-5}$ m/s (n=3), resulting in an estimated flow rate of $4.8 \pm 0.7 \times 10^{-14}$ m/s (i.e., 48 ± 7 pL/s), based on eq. 3.

$$Q = Av \quad (\text{eq. 3})$$

Where Q is the flow rate; A is the cross-sectional area of the capillary (i.e., 7.1×10^{-10} m²); and v is the average laminar flow velocity.

As pointed out by this Reviewer, theoretical and experimental approaches may result in inaccuracies and discrepancies. Consequently, we provided the theoretical and experimental flow rate values in the revised manuscript (page 19). To address the Reviewer's question, additional

experimental details were also provided in the Methods section of the revised manuscript (page 19 and Supplementary Methods).

2) Page 6, line 247 - I would like to know why a short plug of water was injected after introducing the sample, before injecting the PNGase F solution. I would appreciate a brief comment on this in the text.

In this study, a stacking strategy was used to increase the peak intensities and sharpen the detected peaks, for optimized detection and separation of the glycans, after their in-capillary release with PNGase F. To enable this strategy, we tried to highly decrease the concentrations of salts present in 1- the cell culture medium used to inject the mammalian cells inside the capillary, and 2- the commercial PNGase F solution. Therefore, the commercial PNGase F enzyme was diluted 7-fold in water, and the cells were resuspended in 1 mM ammonium acetate pH 6.7, immediately prior to their loading into the CE capillary. We noticed that this low molarity (i.e., 1 mM) of ammonium acetate buffer enabled the preservation of the cell integrity during the injection inside the capillary as well as during the deglycosylation step for N-glycan release (see pages 8 and 20). Nevertheless, to decrease the salt concentrations in the sample zone further, a short water plug was injected after the cell loading in the capillary, before the injection of a second PNGase F plug. These conditions helped the formation of a low conductivity zone after the mixing of the sample plug, water plug, and PNGase F plugs, following the application of two voltage pulses. After the switching of the CE voltage, the glycans released in a low conductivity zone migrate fast and are stacked as a sharp band at the boundaries between the sample and BGE zones. The short water plug injected after the sample injection was also performed to clean the CE capillary inlet before the second PNGase F plug injection in order to avoid potential cross-contaminations between the sample and endoglycosidase solutions. According to the Reviewer's comment, the supporting information was expanded with additional details (see page 20 and Supplementary Methods).

Overall, I can only congratulate the authors of this exciting work.
Tomas Krizek

The authors would like to thank Dr. Krizek for his great support and enthusiasm!

Reviewer #3:

In their study Marie et al. present the first unbiased and label-free glycomics study on single cell level using CE-MS. They identify around 15 and 60 N-glycans from the cell surface of HeLa and U87 cells respectively while maintaining the integrity of the cells. This is of high potential interest as it could allow the combined analysis with the proteome of the very same cell in future. The authors further apply their sensitive method to ultra-low input amounts of extracellular vesicles and plasma isolates.

The technical challenges taken here can also be of help for other ultra-low input applications, not only limited to glycomics studies. They will likely be further improved with newer generation MS instrumentation. That said, this reviewer is convinced that this pioneering SCG study will be of high interest to the community, however some concerns and questions should be addressed as detailed below.

Comments:

1- In this study IgM and IgG were used as model glycoproteins. At the lower end 0.1 ng IgM and 0.5 ng IgG were used. I was wondering why different amounts of protein were selected for the comparison of IgM and IgG. The method section states to this that those inputs correspond to 60 and 50 pL of commercial human serum respectively (again not the same amount, but this might of course vary dependent on the source).

We thank this Reviewer for their valid question, which closely aligns with question 3 raised by Reviewer 1. Please see above the answer we provided to Reviewer 1 to a similar question. Additional details were provided in the Supplementary Note 1 of the revised manuscript (page 4).

2- As described, cells were squeezed in the capillary when the cell-size was close to its inner diameter and this effect was used to facilitate cell stacking.

- What was the exact inner diameter of the capillary, and did it correspond to the average size of a HeLa/U87?

The inner diameter of the capillaries used in our study was 30 μm . Therefore, in our experiments, the capillary diameter was larger than the reported average diameters of HeLa and U87 mammalian cells (PMID: 10933952). According to our measurements (see Methods section page 19), the average diameters of representative HeLa and U87 single cells were $21.8 \pm 4.9 \mu\text{m}$ and $26.1 \pm 6.6 \mu\text{m}$, respectively (based on selected populations of 564 HeLa and 543 U87 cells). Since the cell lines used in this study were cultured under the same conditions, the size distributions of HeLa and U87 cells were considered as Gaussian distributions. Therefore, HeLa and U87 cell populations were divided into three subpopulations equal in size, namely, "smaller", "medium", and "larger" cells. HeLa cells with a diameter $\leq 19.6 \mu\text{m}$ and a diameter $\geq 24.0 \mu\text{m}$ were considered as smaller and larger HeLa cells, respectively (HeLa cells with a diameter in the range 19.6-24.0 μm were considered as medium size cells). For U87 cells, the thresholds to determine smaller and larger U87 cells were 23.2 μm and 29.0 μm , respectively.

- Did you observe problems with stacking for smaller cells, or were they still sticky?

As highlighted in page 7 of the revised manuscript, upon introduction into the capillary, the cells exhibited a tendency to weakly and transiently adhere to the capillary surface. This phenomenon of cell immobilization is likely attributable to the formation of hydrogen bonds and van der Waals interactions between the silanol groups of the bare fused silica capillary surface and numerous chemical groups present on the cell surface. This phenomenon was exploited to stack smaller cells (with a diameter $\leq 19.6 \mu\text{m}$ (HeLa) and $\leq 23.2 \mu\text{m}$ (U87), according to our measurements) during the cell loading process. The "stickiness" of cells to the capillary wall is expected to be different for different cell types.

- How can the chosen workflow be adapted to other cell-types, especially larger ones (I guess one would need dedicated capillaries for each cell type, but very heterogenous populations might be problematic?)

We thank the Reviewer for bringing up a good point. Our POC experiments were performed with two types of relatively large mammalian cells, with specific cell sizes that fit well with the internal diameter of the selected capillary (i.e., 30 μm). As stressed out above, larger HeLa and U87 cells, which may possess a diameter slightly higher than the diameter of the capillary could still be injected inside the capillary. Cells are indeed deformable and can be squeezed upon entering the open tube of the CE capillary, provided that the hydrodynamic flow created between both ends of the capillary is sufficiently high. Our future investigations will focus on the analysis of other types of cell lines with various sizes. This will allow us to determine which cell lines require the use of capillaries with larger diameters. Nevertheless, we believe that the cell size is not the only criterion when selecting the size of the capillary diameter. Cells exhibit distinct physicochemical properties and cell membrane and surface molecules, which may significantly impact the cell deformability/elasticity and adherence to the capillary wall. For specific cell lines, increasing the hydrodynamic flow rate created to load the cells inside the capillary may be sufficient to inject cells larger than HeLa and U87 cells (this can be achieved by automated injection). We agree that the analysis of heterogeneous populations might be challenging and will require further method development and alternative strategies. Our POC study was focused on the analysis of single cells and small numbers (e.g., tens) of cells (HeLa and U87). Nevertheless, we believe that the developed workflow is well adapted to the analysis of small populations of heterogenous cells, including mixed cell line populations, provided that the cells can be injected into the CE capillary.

To address the points raised in question 2 of this Reviewer, additional details were provided in the revised manuscript (pages 7-8, and 19).

3- Figure 3A-C): The profile of 1x HeLa looks very different to 5x or 10x cells and looks way more symmetrical. Do you have an idea why and have you compared profiles from single cells between replicates as well?

The streaks of ions that are detected at ~ 36 - 37 min in Figure 3A and ~ 34 min in Figure 3CD do not correspond to glycan entities. Most probably, these streaks correspond to narrow bands of salts or other low molecular mass species contained in the sample matrices. Besides, the same kind of streaks are also detected in the CE-MS analysis of U87 cells, cell suspension medium, and water blank, but at lower intensities. Very faint bands are detected in the water blank, which confirms our hypothesis. The ion density maps recorded in five CE-MS analyses of single cells showed similar and reproducible CE-MS migration patterns. We clarified these experimental results in page 26 of the revised manuscript.

4- In Figure S3A the variance of raw abundance of glycans from single HeLa seems even lower than that of bulk replicates – how does this agree with the described (and expected) higher variance of glycan abundance in single cells (lines 615-618)?

Due to cell-to-cell heterogeneity and cell size variations, high variations in the absolute abundances of glycans detected in single cells were expected. We noticed that these trends were less pronounced in the CE-MS analyses of small populations of cells (i.e., 5-10 cells). Based on our experimental results (Supplementary Figure 3A), the relative standard deviation (RSD) of the summed raw abundances of eight representative N-glycans was 103% in the measurements of

single cells, but was significantly lower, i.e., 55% and 64%, in the measurements of five and ~ten cells, respectively, based on peak area measurements (considering the variation in the number of cells injected in the CE-MS analysis of ~ten cells (see Methods section), the RSD in glycan abundances should be <64% for ~ten cells). Interestingly, as shown in Supplementary Figure 3B, a linear relationship was demonstrated between the injected cell numbers and the total cellular glycan amounts, based on peak area measurements. Clarifying details were provided as Supplementary Note 2 in the revised manuscript (see page 9).

5- As already highlighted by the authors themselves, it is surprising that the U87 cells yielded much more glycosylation than expected when comparing them to HeLa even when considering their size difference. It might be important for future biological studies to ensure that all or at least close to all glycans are successfully released upon digestion to get a proper fingerprint of each cell. As already suggested in line 683-685 of the manuscript this might not be the case for HeLa. To check one could repeat the analysis with an alternative glycosidase or chemically release glycans instead (e.g. using NaOCl, doi: 10.7171/jbt.19-3004-001).

We thank the Reviewer for their helpful comment and suggestion. As highlighted on page 11 of the revised manuscript, several factors may explain the different numbers of N-glycans detected at the surface of HeLa and U87 single cells, including unique molecular features, intrinsic cell morphology, and structural characteristics specific to each cell line. Our study was focused on the analysis of N-glycans. Therefore, PNGase F was selected in our POC experiments because this enzyme was reported as the most effective enzyme for the release of intact and native N-linked glycans. Alternative endoglycosidases (e.g., PNGase A, Endo H, or Endo F, and new generations of recombinant PNGases) cleave only specific types of glycans. For instance, PNGase A is unable to release complex sialylated glycans. Also, Endo H and Endo F do not have the ability to release intact N-glycans (i.e., these enzymes cleave a glycosidic bond between two monosaccharides and not the amide bond between the innermost GlcNAc residue and the asparagine residue of the protein backbone). The use of chemical agents could, indeed, be an alternative option for N-glycan release, which may work better for analytes other than intact cells. In the paper mentioned by this Reviewer (PMID: 31598098), it was demonstrated that sodium hypochlorite (NaOCl) releases N-glycans less efficiently than PNGase F and that NaOCl could also induce substantial glycan degradation. Therefore, we do not think that NaOCl might be used for the release of N-glycans from the surface of single cells. Nevertheless, we think that the question raised by this Reviewer deserves further investigation in the future. Additional comments were introduced on pages 15-16 of the revised manuscript.

6- This work is a very exciting first step into performing unbiased glycomics at the single cell level and will be important for future biological studies as well. With this in mind, I was wondering if you can comment on scalability? Would it be possible to do this in a high throughput manner i.e. 100 or more cells in a reasonable time (can it be automatized, how many cells can be stacked into one capillary,...?).

Great points! Future investigations will allow us to assess the scalability and throughput of our developed CE-MS-based technology. Automated injection was used for the injection of ~10 cells (see Methods section page 21) and can be used for the injection of higher numbers of cells. We expect that the N-glycan profiling of ~50 cells (or even more) can be achieved using our developed

workflow. Automated injections decrease the total analysis time down to ~2.5 h. The scalability perspective was introduced in page 15 of the revised manuscript. We also plan to investigate further possible strategies to increase the throughput in analyzing single cells, including the multi-segmented injection approach, which is currently technically challenging to implement.

7- I tried to check the data on GlycoPOST but got a “No object registered” prompt when searching for the dataset GPST000378. Is it publicly available or do I need a password (ahead of publication)?

The raw data generated in this study have been deposited in GlycoPOST (GPST000378 and GPST000380). During the manuscript review cycle, the deposited data are accessible with the passwords provided below.

<https://glycopost.glycosmos.org/preview/20568383116532e243a77e7> (Pin code 7278)

<https://glycopost.glycosmos.org/preview/15159624726545509d36266> (Pin code 5722)

Minor Comments:

8- Line418 seems to be grammatically wrong/ a typo: Suggestion: “We noticed that the development and optimization of the workflow as well as its adoption to...”

Addressed in the revised manuscript (page 5).

9- Fig 3H-K : The annotations are very hard to read, upon zooming, the quality is not good enough to properly read the numbers in the Venn diagrams. Maybe those panels can be moved to a separate figure or at least somehow enlarged.

Addressed. We increased the size font of the numbers displayed in Figure 3H-K.

10- Line 624: I guess this should be referenced to Figure 4I (not 4E).

Thank you for bringing this to our attention! Figure 4I was indeed the figure we wanted to refer to (see page 9).

11- General: When reading the manuscript, it seems there is a lot of jumping up and down between figures which lowers smooth readability. I'd recommend restructuring a bit to ease readability. (E.g., figure 4J and K is mentioned in the text the first time after figure 6 description and could be moved down.)

In our revised manuscript, we deemed it relevant to gather in Figure 4 all the panels related to the fractional distributions of sialylated and fucosylated glycans identified in HeLa and U87 cells, with and without LPS treatment. This provides an overview, which facilitates the comparative analysis between the different cell lines and their altered glycosylation profiles. In the same way, the MS² spectra acquired in CE-MS² analysis of HeLa and U87 cells were all displayed in Figure 6.

Reviewers' Comments:

Reviewer #1:

Remarks to the Author:

This manuscript addresses a clear need for methods for analysis of glycans from single cell biological samples. In previously published work, the authors employed a manual hydrodynamic cell loading method for analysis of 1-10 mammalian cells. In the present work, they further optimized this manual method. For CE with optical detection, it is necessary to derivatize glycan samples using a chromophore or fluorophore. Many MS-based methods also employ such derivatization because it enhances ionization signal strength. But such derivatization is not practical when manipulating minute volumes and it is significant that the authors have shown ability to detect underivatized released glycan quantities on the scale of single mammalian cells. Overall, the results are presented qualitatively and lack assignment criteria and confidence measures. Without this information it is difficult for the reader to judge the effectiveness of the manual injection method. There is a lack of glycan abundance reproducibility data that would lend confidence to the usefulness of the new technology. The method is not mature enough to be used for reaching biological conclusions.

Abstract: State more precisely the ng range of protein required for the analysis from the biological samples.

P. 3. It is not clear what is meant by "sequencing-based glycomic profiling of single cells". Since this refers to lectin-binding signals, the use of "sequencing" may confuse the reader.

P. 4. It is not clear against what procedure "The vastly simplified in-capillary sample preparation approach" is compared. Is this an improvement over the authors' previous publication? Or a different publication? Also, does "the offline approach" refer to the authors' previous publication?

P.4. Avoid vague statements of performance. Give ranges for the performance of the method described here. Rather than using "sub-ng-levels" state the range in pg analyzed. State the range in pL volumes of total plasma. Does this refer to 1-10 pL? or 10-100 pL? or to something else?

P. 4. The observation of changes in N-glycan analysis resulting from "biochemical stimulation" is a weak validation for the new method. It would be better to use a perturbation method that produces a well benchmarked change that would lend confidence to the new method.

Fig. 2. Discuss criteria for inclusion of glycans in the assignment lists. Give confidence metrics for the assignments. Provide data on the glycan abundance reproducibility and coefficients of variation for the sample shown in this and all figures.

Fig. 5. It is over-reaching to assign glycan topology from the tandem mass spectra. Label the tandem mass spectra with glycan compositions. If the authors wish to assume the glycan topology from some literature report, they should include this in the discussion.

Fig. 6. Discuss the steps taken to understand sources of bias and control experimental error in this experiment. Show as supplementary information the order of analysis of the single cell samples. Describe the criteria for inclusion of glycans in assignment lists for this experiment. Discuss for single cells how to differentiate biological change versus technical variation. Demonstrate convincingly that the results show change in glycan abundances that are due to perturbation rather than experimental variation.

Supplementary Fig. 3 shows glycan abundance variation is high for these samples. With such variability, how can confident conclusions be reached regarding significance of abundance differences in different biological samples? What is the effect size required to quantify the glycan abundance difference between two single cells?

Reviewer #3:

Remarks to the Author:

The authors addressed all questions of the reviewers and improved the clarity and quality of the manuscript. The study will likely be of high interest and value to the community and includes a novel approach for N-glycan profiling of single cells with great potential!

I can now recommend the manuscript for publication and add just some minor comments I have to the revised work.

- Supplemental Tables have been added and are of great support for the reader. The only remark I'd have is to add something like a table annotation or heading explaining what is shown (i.e.,

mean # of glycans identified, abundance, etc.).

- I'd like to thank the authors for their detailed explanation on question 3/reviewer 3 regarding the ion intensity maps of Figure 3A-C. My question might be naïve, as I have expertise in proteomics using LC-MS with low experience in CE-MS, but I am still wondering why there are strong signals in panel A (single HeLa) at 35-38min and why those are missing for panel B-F (more cells or different cell type). If those signals originate from salt, as suggested in the author's response, why is the salt content different? The authors further mention that clarifying details have been added to page 26. On page 26, figure 3 is shown and there the #of replicates used to yield the shown mean #of identified glycans was added but I couldn't find additional comments on the ion intensity maps. Please ignore this comment in case the seen ion mobility maps are already as expected for someone more into CE-MS and hence do not require additional discussion.

- The same applies to my last comment: The authors explain, that the SD of summed raw abundances of eight representative N-glycans was 103% in the measurements of single cells, but was significantly lower, i.e., 55% and 64%, in the measurements of five and ~ten cells, respectively. I fully agree with the authors that this is in line with expectations. What still seems surprising to me is that it seems not to align with the data shown in supplemental figure 3 A and B. There the summed raw abundances of 1-10 cells are shown and the error bars (representing SD) are clearly higher (not lower). Coming from single cell proteomics, we usually see high error bars for protein quantities of replicates from single cells and lower error bars for replicate measurements of higher cell numbers, i.e., from 10 cells. Again, sorry in case I misinterpreted the graph.

Point-by-point responses to Reviewers' comments

Title: In-capillary sample processing coupled to label-free capillary electrophoresis-mass spectrometry to decipher the native N-glycome of single mammalian cells and ng-level blood isolates

Authors: Anne-Lise Marie, Yunfan Gao, and Alexander R. Ivanov

The authors thank the Reviewers for their thorough examination of our manuscript and for providing helpful recommendations. We addressed the last concerns and comments made by the Reviewers in full. We strongly believe that the second revised version of the manuscript was significantly improved with the substantial and carefully crafted changes that we made based on the Reviewers' suggestions.

Please see below our point-by-point responses in blue font.

Reviewer #1:

This manuscript addresses a clear need for methods for analysis of glycans from single cell biological samples. In previously published work, the authors employed a manual hydrodynamic cell loading method for analysis of 1-10 mammalian cells. In the present work, they further optimized this manual method. For CE with optical detection, it is necessary to derivatize glycan samples using a chromophore or fluorophore. Many MS-based methods also employ such derivatization because it enhances ionization signal strength. But such derivatization is not practical when manipulating minute volumes and it is significant that the authors have shown ability to detect underivatized released glycan quantities on the scale of single mammalian cells. Overall, the results are presented qualitatively and lack assignment criteria and confidence measures. Without this information it is difficult for the reader to judge the effectiveness of the manual injection method. There is a lack of glycan abundance reproducibility data that would lend confidence to the usefulness of the new technology. The method is not mature enough to be used for reaching biological conclusions.

We acknowledge Reviewer 1 for their thorough examination of our manuscript. We hope that the additional clarifications and supplementary data provided in the revised version of our manuscript will be helpful. In the revised manuscript, a substantial part of the Results section describes quantitative results, with a thorough comparison of quantitative profiles of N-glycans detected in the analyzed biological samples, based on peak intensity/area measurements (pages 7, 9, 10, and 12-14). Moreover, numerous revised figures and tables display the abundances of glycans detected in the biological samples (Figure 2J, Figure 4I, Figures 6A and 6C, Supplementary Figures 3A-D, Supplementary Figures 5C-F, and Supplementary Tables 2 and 5). The relative standard deviations (RSDs) of glycan abundances were also provided in the Supplementary Materials. We, therefore, think that our manuscript provides a fine and appropriate balance between the amounts of the

included qualitative and quantitative results. In addition, our proof-of-concept (POC) experiments were systematically validated by a series of statistical tests (see Supplementary Table 4 of the revised manuscript). Please see below our point-by-point responses.

1- Abstract: State more precisely the ng range of protein required for the analysis from the biological samples.

Addressed on page 1 of the revised manuscript.

2- P. 3. It is not clear what is meant by “sequencing-based glycomic profiling of single cells”. Since this refers to lectin-binding signals, the use of “sequencing” may confuse the reader.

We thank the Reviewer for bringing this to our attention. Additional clarification was provided in the revised manuscript (page 3).

3- P. 4. It is not clear against what procedure “The vastly simplified in-capillary sample preparation approach” is compared. Is this an improvement over the authors’ previous publication? Or a different publication? Also, does “the offline approach” refer to the authors' previous publication?

We thank the Reviewer for bringing up this question. In the field of glycomic analysis, the most commonly used approaches involve glycan derivatization for improved detectability and separation of glycans with CE (or other separation methods, like liquid chromatography) and/or MS-based techniques. As stressed out in our recently published work (PMID: 36959283), label-free CE-MS analysis of enzymatically released N-glycans simplifies and shortens the analytical workflow, while highly reducing sample losses during sample preparation and cleanup steps associated with any labeling procedure. In the presented study, the online coupling of in-capillary sample preparation with CE-MS analysis even further simplifies the analytical workflow with the elimination of sample handling and transfer steps associated with any offline procedure. Therefore, the term “vastly simplified” refers not only to our previously published work (PMID: 36959283) but also to other reported techniques focused on glycomic profiling of minute amounts of biological samples using an offline approach and/or a derivatization strategy (PMIDs: 18973241, 29528644, and 34505713). For the sake of clarity, additional details were provided in the revised manuscript (page 4).

4- P.4. Avoid vague statements of performance. Give ranges for the performance of the method described here. Rather than using “sub-ng-levels” state the range in pg analyzed. State the range in pL volumes of total plasma. Does this refer to 1-10 pL? or 10-100 pL? or to something else?

As suggested by the Reviewer, in the revised manuscript, we stated more precisely the mass/volume ranges for the analyzed blood-derived isolates (page 4).

5- P. 4. The observation of changes in N-glycan analysis resulting from “biochemical stimulation” is a weak validation for the new method. It would be better to use a perturbation method that produces a well benchmarked change that would lend confidence to the new method.

To the best of our knowledge, there are currently no clear-cut benchmarking methods that involve precise manipulation of induced glycome changes (e.g., genetic perturbation methods or biochemical/biophysical techniques) leading to predictable, highly similar, and reproducible glycan profile alterations across various cell types that can be used at a single-cell level analysis. However, we used well-established treatment models that induced glycan profile changes that were characterized in detail in recent publications. As stressed in the revised manuscript (page 13), several studies reported specifically altered glycan sialylation or fucosylation profiles upon lipopolysaccharide (LPS) treatment of a variety of mammalian cells/tissues, including both immune and non-immune cells (PMIDs: 23211310, 31445704, and 33993882). We, therefore, deemed the selection of the model system and LPS for our POC study. Our experimental results were systematically validated by statistical tests (see Supplementary Table 4 in the revised manuscript). In addition, we are convinced that chemical and biological treatments of cells with therapeutic drugs and drug candidates that target the level and specific types of cellular glycosylation can induce significant glycome alterations that could be detected at the single-cell level using our developed workflow. Therefore, we think that biochemical stimulation of cells previously reported in the literature, including LPS treatment, can serve as an efficient and very informative method in single-cell glycomic (SCG) profiling studies focused on disease biomarker and drug discoveries. Additional clarifications were provided on page 13 of the revised manuscript.

6- Fig. 2. Discuss criteria for inclusion of glycans in the assignment lists. Give confidence metrics for the assignments. Provide data on the glycan abundance reproducibility and coefficients of variation for the sample shown in this and all figures.

We thank the Reviewer for raising these critical points. In our study, the assignment of glycans was based on CE-MS data processing using GlycReSoft software (Boston University, MA) (PMIDs: 29790907 and 32223173). The search parameters (mass error tolerance, background reduction factor, scoring threshold, adduct ions, etc.) were described in the Methods section of the revised manuscript and in our referenced recently published papers (PMIDs: 36959283 and 33433994), as indicated on page 23 of the revised manuscript. According to the software developers' description (PMID: 29790907), GlycReSoft uses a composite summarization score, which takes into account metrics including, inter alia, chromatographic peak shapes, ion charge state distributions, isotopic pattern consistency, adduct frequency, and the time gap between single-stage MS observations (for missing peaks and interference detection) to distinguish the observed features from MS background noise. For the assignment of glycan compositions to the chromatographic features, GlycReSoft utilizes the biosynthetic network relationship among glycan compositions and their neighboring sub-types. A Laplacian regularization algorithm is applied to combine the observed score and the glycan network graph topology to generate a smoothed score for the final glycan composition assignment (see also our recent paper, where we summarized the scoring process applied in GlycReSoft (PMID: 33433994)). In the presented here study, the glycan identification analysis of the CE-MS data was conducted using database searches against an in-house built mammalian database encompassing 27,335 N-glycan compositions (see page 23). As additional verification of the plausible glycan composition identifications made using GlycReSoft, several supplementary levels of manual data examination were applied according to our recent studies (PMIDs: 36959283 and 33433994). In brief, this verification included 1. Predictable trends in CE-MS migration patterns with respect to glycan composition, net charge, and molecular mass; 2. Charge state and isotopic distributions characteristic to glycan ions; 3. Detection of neutral

losses of monosaccharides (e.g., hexose and N-acetyl-hexosamine); and 4. Manual examination of CE-MS² data for low intensity parent ions (see page 23). For the sake of clarity, additional information was provided in the revised manuscript (page 23).

As suggested by this Reviewer, the RSDs of glycan abundances were also provided in the Supplementary Materials and Supplementary Figures 3A-D. The Results section was also expanded with additional experimental results on glycan abundance reproducibility (pages 9-10). See also our response to point 9 below.

7- Fig. 5. It is over-reaching to assign glycan topology from the tandem mass spectra. Label the tandem mass spectra with glycan compositions. If the authors wish to assume the glycan topology from some literature report, they should include this in the discussion.

The structural characterization of the glycans detected in the biological samples analyzed in this study was based on highly informative CE-MS² spectra acquired in negative ion mode using the developed label-free workflow. As highlighted in previous studies reported by our and other groups (PMIDs: 36959283, 11055725, 12322961, 26842584, 18327885, 8914337, and 22120881 - to quote just a few of them), tandem MS-based fragmentation of non-labeled glycans in negative ion mode provides extensive and highly abundant cross-ring cleavages that enable unambiguous and accurate characterization of glycan structural features such as monosaccharide composition, antenna branching, location of fucose residues, and nature of SiA linkages. In the presented study, no glycan topology was assumed from the literature data. Previously published reports (including the aforementioned articles) helped us elucidate the CE-MS² fragmentation patterns of the detected glycans. In the revised manuscript, the glycan compositions of all the glycans characterized by CE-MS² were described in Supplementary Tables 3, 5 and 6. Based on the Reviewer's suggestion, the composition of the glycans presented in Figure 5 were added in the caption of Figure 5 (page 30), and additional references were included (pages 10 and 13) in the revised manuscript.

8- Fig. 6. Discuss the steps taken to understand sources of bias and control experimental error in this experiment. Show as supplementary information the order of analysis of the single cell samples. Describe the criteria for inclusion of glycans in assignment lists for this experiment. Discuss for single cells how to differentiate biological change versus technical variation. Demonstrate convincingly that the results show change in glycan abundances that are due to perturbation rather than experimental variation.

We thank the Reviewer for raising critical questions. In our POC study, the reproducibility of the method in the CE-MS analysis of single cells was assessed using at least five repetitive experiments. Two different mammalian cell types (HeLa and U87 cells) were selected for the initial performance evaluation of the developed single-cell glycomic approach. To minimize any potential analytical biases, in particular the levels of carryover derived from the analysis of preceding mammalian cells, and provide accurate and reliable results, thorough and careful rinses of the capillary and control analyses with a water blank sample were systematically performed between runs, as indicated on pages 9, and 22, and shown in Figure 3EG and Supplementary Table 1 of the revised manuscript. During the method development and optimization stage of this study, CE-MS analyses of single mammalian cells were carried out through many days and with different CE capillaries without observing major differences in performance. Unsuccessful measurements,

caused, for example, by hardware malfunction (mechanical capillary or emitter damage), were excluded from this POC study. Therefore, we determined that the glycome alterations that we observed at the single-cell level between different cell types and between untreated and LPS-treated cells were not derived from analytical biases or other sources of experimental/technical variations. Finally, as explained above in the response to point 6, the assignment of glycans was based on CE-MS data processing using not only the GlycReSoft software but also on supplementary levels of careful manual data examination. For all these reasons, we think that the results provided in our POC study, which were validated by a series of statistical tests, are of high confidence. Additional clarifications were provided in the revised manuscript (pages 21-23). Because of the cell-to-cell heterogeneity, it is highly challenging to directly assess the level of analytical variation in single-cell analysis. Nevertheless, with the statistical tests we performed, technical variations together with cell-to-cell variations were taken into account when comparing the LPS-treated and untreated cells.

9- Supplementary Fig. 3 shows glycan abundance variation is high for these samples. With such variability, how can confident conclusions be reached regarding significance of abundance differences in different biological samples? What is the effect size required to quantify the glycan abundance difference between two single cells?

As emphasized on page 9 of the revised manuscript, due to cell-to-cell heterogeneity (arising from molecular heterogeneity, and cell size, cell surface area, and cell cycle state variability, inter alia), high variations in the absolute (i.e., raw) abundances of glycans detected in single cells were observed. As expected, these trends were less pronounced in the CE-MS analyses of small populations of cells (i.e., 5-10 cells), as shown in Supplementary Figure 3 and described in Supplementary Note 2 of the revised manuscript. As shown in Supplementary Figure 3B as an illustrative example, the normalization of glycan abundances (i.e., each glycan abundance was normalized with respect to the summed abundances of all the selected N-glycans, based on peak area measurements) resulted in significantly lower RSDs of quantitative values, in comparison to the RSDs of absolute abundances of the same selected glycans (Supplementary Figure 3A). Based on the normalized abundances of detected glycans and a series of statistical tests (see Supplementary Table 4), our POC experiments allowed us to show significant differences between HeLa and U87 cell N-glycomes. We would like to emphasize that this POC study explores uncharted territories of MS-based single-cell glycomic profiling where the intercellular alterations of glycomes are hitherto unknown. The RSDs of glycan abundances measured in our POC SCG experiments are similar to the reported RSD values for single-cell proteomic abundance measurements and substantially more accurate than the RSD values recently reported in single-cell transcriptomic studies (i.e., a higher variability of abundance measurements was demonstrated for single-cell RNA measurements compared to single-cell proteomic data, see PMID: 33504367). For instance, in single-cell transcriptomic studies, median RSDs of quantitative measurements might be well over 600%, as reported in some of the recent scRNA seq studies (PMIDs: 30283141, 31345182, and 24141493). However, a clear separation of the cell types based on UMAP plots were still observed despite such high variability. In the case of single-cell transcriptomics (scRNA seq), the RSDs for quantitative profiling measurements can also be increased during the amplification process beyond the RSD levels commonly observed in single-cell proteomic and glycomic measurements. The current implementations of the scRNA seq technique still demonstrate high variations in single-cell analysis. In our POC SCG study, the median RSDs of

non-normalized and normalized glycan abundances of a set of selected highly representative N-glycans were 106% and 28%, respectively (see Figure 3AB). These results align well with the RSDs of protein abundances reported in single-cell proteomic studies (e.g., median RSDs of protein abundances accounting for over 30% and up to 90% were reported in proteomic analysis of single lung cells. Similar or even higher median RSDs were reported for various mammalian cells in other studies as well. (PMIDs: 29797682, 34967612, and 37737208)). Expectedly, larger-scale studies involving a larger variety of cell lines and higher numbers of experimental replicates (i.e., repetitive CE-MS analyses of single cells or tiny amounts of cells) will be required in the future to more accurately determine the RSDs of glycan abundances for the glycans detected in single cells and small populations of cells. Such large-scale studies were not the goal of our POC experiments. In the revised manuscript, we expanded the Results and Discussion sections with clarifying details and additional references to the literature data, as well as Supplementary Figures (e.g., Supplementary Figure 3A and 3B) (see pages 9-10, and 15-16).

The high cellular and molecular heterogeneity of tissues and cultured cells is the reason for developing analytical techniques capable of detecting and mapping heterogeneities between individual cells. Single-cell transcriptomics and proteomics studies have identified subpopulations of cell lines (with significant intercellular homogeneity) in tumor tissues and cell cultures, based, for example, on specific patterns of quantitative RNA or cell surface protein marker profiles (see, e.g., PMIDs: 20964822 and 35858333). Yet, such studies required the analysis of hundreds or thousands of single cells to profile cell subpopulations. The primary purpose of our POC study was to demonstrate the capability of the developed CE-MS-based workflow to detect and quantify glycans at the single-cell level sensitively and accurately. Future investigations in SCG most probably will result in the identification of subpopulations of cell lines, based on specific cell surface glycan markers, but such investigations will require large-scale studies.

Reviewer #3:

The authors addressed all questions of the reviewers and improved the clarity and quality of the manuscript. The study will likely be of high interest and value to the community and includes a novel approach for N-glycan profiling of single cells with great potential! I can now recommend the manuscript for publication and add just some minor comments I have to the revised work.

The authors acknowledge the Reviewer for their positive and enthusiastic feedback, and for providing thoughtful and helpful comments. We addressed the latest concerns raised by this Reviewer in the revised version of our manuscript.

1- Supplemental Tables have been added and are of great support for the reader. The only remark I'd have is to add something like a table annotation or heading explaining what is shown (i.e., mean # of glycans identified, abundance, etc.).

Addressed in the Supplementary Materials of the revised manuscript.

2- I'd like to thank the authors for their detailed explanation on question 3/reviewer 3 regarding the ion intensity maps of Figure 3A-C. My question might be naïve, as I have expertise in proteomics using LC-MS with low experience in CE-MS, but I am still wondering why there are

strong signals in panel A (single HeLa) at 35-38min and why those are missing for panel B-F (more cells or different cell type). If those signals originate from salt, as suggested in the author's response, why is the salt content different? The authors further mention that clarifying details have been added to page 26. On page 26, figure 3 is shown and there the #of replicates used to yield the shown mean #of identified glycans was added but I couldn't find additional comments on the ion intensity maps. Please ignore this comment in case the seen ion mobility maps are already as expected for someone more into CE-MS and hence do not require additional discussion.

We thank the Reviewer for this valuable and thoughtful comment. It is indeed quite usual to detect streaks of ions in CE-MS analysis of glycans (or other molecular species, such as proteins) released from biological samples. It is rather difficult to determine the nature of the molecular species (detected mainly as singly-charged ions) corresponding to these ion streaks. They are most probably derived from the sample matrices, which are particularly complex in the case of heterogeneous biological specimens. Due to the relatively high amounts of salts or salt-associated molecular species in the analyzed biological samples, we assumed that the detected ion streaks corresponded to narrow bands of salts or other low molecular mass species contained in the sample matrices. The ion streaks observed in the ion density maps acquired in our CE-MS analyses may be detected within slightly different migration time windows and at different intensities, depending on the nature and amounts of injected samples. The CE-MS analyses of single cells result in additional sources of variability, and, for such analyses, the matrix effect may be amplified due to the extremely low amounts of injected biological material. This hypothesis may explain a distinct CE-MS migration pattern for the analyzed single cells, in comparison to the CE-MS analyses of small populations of cells. Based on the Reviewer's comment, additional clarifications were provided in Supplementary Note 2 of the revised manuscript (see page 9) based on the best of our current knowledge.

3- The same applies to my last comment: The authors explain, that the SD of summed raw abundances of eight representative N-glycans was 103% in the measurements of single cells, but was significantly lower, i.e., 55% and 64%, in the measurements of five and ~ten cells, respectively. I fully agree with the authors that this is in line with expectations. What still seems surprising to me is that it seems not to align with the data shown in supplemental figure 3 A and B. There the summed raw abundances of 1-10 cells are shown and the error bars (representing SD) are clearly higher (not lower). Coming from single cell proteomics, we usually see high error bars for protein quantities of replicates from single cells and lower error bars for replicate measurements of higher cell numbers, i.e., from 10 cells. Again, sorry in case I misinterpreted the graph.

We thank the Reviewer for their valuable comment. We agree that Supplementary Figures 3A and 3B (renamed 3C and 3D in the revised manuscript) may result in misinterpretation of the standard deviations (SDs) and relative standard deviations (RSDs) in the measurements of glycan abundances for the eight selected representative glycans, due to graphical representation. In the above-mentioned Supplementary Figure 3, the SDs (corresponding to the error bars) were 2.5E6, 8.8E6, and 1.8E7, for one, five, and ~ten cells, respectively. The determined RSDs were, therefore, 103%, 55%, and 64%, for one, five, and ~ten cells, respectively (the means of the summed raw abundances being 2.4E6, 1.6E7, and 2.8E7, respectively). For the sake of clarity, we included in the revised manuscript the values corresponding to the means, SDs, and RSDs of glycan abundances in Supplementary Figures 3C and 3D.

Reviewers' Comments:

Reviewer #1:

Remarks to the Author:

The authors have addressed my concerns appropriately.